# Uneven consequences of global climate mitigation pathways on regional water quality in the 21st century

Minjin Lee [1] ✉, Charles A. Stock [2], Elena Shevliakova [2], Sergey Malyshev [2], Maureen Beaudor [3] & Nicolas Vuichard[4]

Future socioeconomic climate pathways have regional water-quality consequences whose severity and equity have not yet been fully understood across geographic and economic spectra. We use a process-based, terrestrial-freshwater ecosystem model to project 21st-century river nitrogen loads under these pathways. We find that fertilizer usage is the primary determinant of future river nitrogen loads, changing precipitation and warming have limited impacts, and $CO_2$ fertilization-induced vegetation growth enhancement leads to modest load reductions. Fertilizer applications to produce bioenergy in climate mitigation scenarios cause larger load increases than in the highest emission scenario. Loads generally increase in low-income regions, yet remain stable or decrease in high-income regions where agricultural advances, low food and feed production and waste, and/or well-enforced air pollution policies balance biofuel-associated fertilizer burdens. Consideration of biofuel production options with low fertilizer demand and rapid transfer of agricultural advances from high- to low-income regions may help avoid inequitable water-quality outcomes from climate mitigation.

Nitrogen (N) fertilizer usage and cultivation-induced biological N fixation (BNF) help feed nearly half the global population[1]. These practices, in addition to fossil fuel burning for energy production, have increased reactive N losses to the environment, causing a cascade of negative impacts on the ecosystem and human health[2,3]. NOx, N₂O, and NH₃ emissions to the atmosphere have contributed to acid rain[4], air pollution[5], stratospheric ozone depletion[6], and the radiative forcing underlying climate change[7]. N fluxes from lands have also impaired freshwater quality[8] and contributed to coastal eutrophication[9], hypoxia[10], and harmful algal blooms[11], putting aquatic resources and the communities that depend upon them at risk[12].

Projected increases in fertilizer use to meet growing demands for food and livestock-based products[1,13] heighten risks to aquatic ecosystems, with agricultural trade raising the prospect that this burden may be inequitably distributed[14]. Furthermore, a new generation of future scenarios (i.e., shared socioeconomic pathways (SSPs)) suggests that bioenergy will play a critical role in climate change mitigation[13], raising concerns about unintended, adverse effects of bioenergy-associated fertilizer applications on aquatic ecosystems[15–17]. Climate change may also impact the delivery of N to rivers and coastal waters through shifting hydroclimate[17,18], while elevated $CO_2$ may exert indirect effects on river N loads through enhanced agricultural productivity and terrestrial N sequestration[19–23]. There is thus a compelling need to understand the combined effects of these concurrently changing drivers on future river N loads and water quality.

Projections to date, however, have focused on a subset of these drivers and the integrated effects of interactions among atmospheric $CO_2$, climate change, and socioeconomic drivers shaping river N loads have not yet been fully understood within a unified process-based framework[17,24–26]. Moreover, previous projections were limited to

[1]Program in Atmospheric and Oceanic Sciences, Princeton University; Princeton, Princeton, NJ, USA. [2]NOAA/Geophysical Fluid Dynamics Laboratory; Princeton, Princeton, NJ, USA. [3]High Meadows Environmental Institute, Princeton University, Princeton, Princeton, NJ, USA. [4]Laboratoire des Sciences du Climat et de l'Environnement (LSCE), CEA–CNRS–UVSQ, Gif-sur-Yvette, France. ✉e-mail: minjinl@princeton.edu

either mid-21st century time horizons before scenario differences are fully realized[24,26] or regional scales[17,25], limiting perspective on the equity of socioeconomic developments and climate policies manifested in regional water quality across geographic and economic development spectra. Furthermore, most prior projections were derived from models that rely heavily on empirical relationships[17,26] or assume that N storage in river basins[26] or parts of river basins[24] is in steady state. Such stationarity assumptions become less reliable for longer time horizons[27], as unprecedented socioeconomic and climate changes push river basins well beyond the scope of prior observations[28].

In this study, we use the process-based, terrestrial-freshwater ecosystem model LM3-TAN[20,29] to project spatially explicit, global river N loads over the 21st century. While LM3-TAN has been implemented at relatively coarse 1-degree resolution and does not resolve river water management (e.g., dams), it simulated global and regional terrestrial and freshwater carbon (C) and N storage and fluxes to the ocean and atmosphere consistently with published synthesis from 28 different studies[18,20]. With necessary updated forcings and minor calibration refinements to make future projections herein (see "Methods", Supplementary Note S1), LM3-TAN is found to accurately capture observed water discharge, nitrate, dissolved inorganic N (DIN), and dissolved organic N (DON) concentrations and loads from 53 large rivers influenced by various climate and socioeconomic conditions, with skill comparable to empirical approaches while maintaining consistency with global soil N storage, terrestrial C storage changes, and river DIN and DON load estimates (Supplementary Note S1, Supplementary Tables S1–S4, Supplementary Fig. S1). LM3-TAN has also simulated realistic long time series N load variability in several river basins[18,29].

We analyze the combined and interactive effects of projected future changes in land use[30], N inputs (i.e., fertilizer applications[30], atmospheric deposition[31], and simulated BNF by LM3-TAN), atmospheric $CO_2$[31,32], and climate (e.g., temperature and precipitation[33,34]) on water N pollution under three Coupled Model Intercomparison Project, Phase 6 (CMIP6) Integrated Assessment Model (IAM) marker scenarios: (i) SSP1-2.6 (Sustainability – Taking the Green Road)[35], (ii) SSP2-4.5 (Middle of the Road)[36], and (iii) SSP5-8.5 (Fossil-fueled Development – Taking the Highway)[37]. These three scenarios span a broad range of climate change and N pollution drivers in the scenario literature[30,38]. We also explore the robustness of our findings to recently developed manure application projections[39-41]. While these projections are not available for SSP1-2.6, scenarios for SSP2-4.5 and SSP5-8.5 and a scenario with the lowest livestock production and manure applications (SSP4-3.4) provide a means of comparing the relative importance of cross-scenario fertilizer and manure application differences. A suite of sensitivity tests is also included to assess uncertainties associated with future climate change and $CO_2$ fertilization (see "Methods").

We explore three primary questions: (i) What will global water N pollution look like under different socioeconomic climate scenarios over the 21st century? (ii) How will water N pollution diverge across low- to high-income (i.e., per capita GDP, pcGDP[42]) regions? and (iii) What are the drivers of the regional divergence?

## Results

Global river dissolved N (DN) loads to the coastal ocean (hereafter "loads") are projected to rise in the 21st century under all three scenarios (Fig. 1a). Increasing trends in river DN loads are aligned with increasing trends in fertilizer applications (Fig. 1c), rather than changes in atmospheric deposition and BNF (Fig. 2a), or precipitation and temperature (Supplementary Figs. S2–S4). That is, both river DN loads and fertilizer applications are highest in SSP2-4.5, intermediate in SSP1-2.6, and lowest in SSP5-8.5, while atmospheric deposition, BNF, precipitation, and temperature are highest

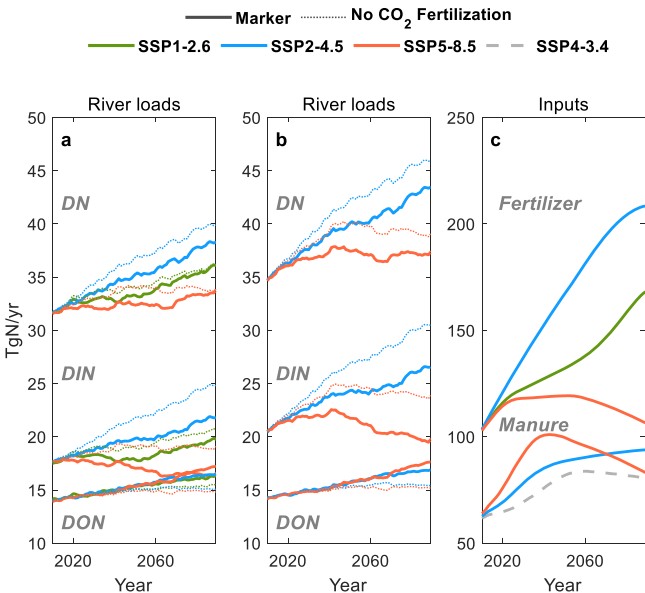

**Fig. 1 | Rising global river N loads and fertilizer applications. a, b** River dissolved inorganic N (DIN), dissolved organic N (DON), and dissolved N (DN, the sum of DIN and DON) loads to the coastal ocean for simulations without manure applications (**a**) and with manure applications (**b**). **c** Fertilizer and manure N applications. All plots show 21-year moving averages from 2000 to 2099. The range across three scenarios (indicated by colors) provides a first-order estimate of uncertainty due to alternative socioeconomic developments and climate policies. The uncertainty of results to future $CO_2$ fertilization effects are demonstrated by comparing the marker scenarios (solid lines) with those that exclude $CO_2$ fertilization (dotted lines). See "Methods" for a detailed description of model simulations. Source data are provided as a Source Data file.

in the highest emission scenario SSP-8.5, followed by SSP2-4.5 and SSP1-2.6.

Differences in manure applications between the scenarios are considerably smaller than the differences in fertilizer applications, even when the "lower bound" scenario is considered (Fig. 1c). Manure applications in SSP5-8.5 are higher than those in SSP2-4.5 until the 2060s, but declines by the end of the 21st century and does not significantly shift the high fertilizer-driven river DN loads in SSP2-4.5 relative to SSP5-8.5 (Fig. 1b). While manure applications for SSP1-2.6 are unavailable, relative patterns in livestock production (Supplementary Fig. S5), a primary determinant of manure production[39-41], suggest that manure applications in SSP1-2.6 lie between the lower bound scenario and SSP2-4.5. This would again yield differences in N inputs due to manure that are far less than cross-scenario fertilizer contrasts. The alignment of trends in river DN loads with those in fertilizer applications also remains when uncertain $CO_2$ fertilization effects (i.e., enhanced vegetation growth due to elevated $CO_2$) are omitted from LM3-TAN (dotted lines in Fig. 1a, b, see "Methods") or when reported extreme limits of climate change uncertainty for SSP5-8.5[43,44] are considered ( + 7.6% precipitation or +5.3 °C temperature by 2080-2099 relative to 1986-2005, Supplementary Figs. S2–S4).

While river DN load trends are aligned with fertilizer application trends, the projected magnitude and range of loads across the scenarios (34-43 TgN yr⁻¹ by the end of the 21st century (2079-2099)) is far smaller than those of fertilizer applications (107–208 TgN yr⁻¹, Fig. 1 and Supplementary Table S5). Other inputs and processes in terrestrial and freshwater ecosystems clearly modulate the imprint of fertilizer applications on river DN loads. Relatively low inputs of BNF and atmospheric deposition (Fig. 2a, b) and high denitrification rates (Fig. 2c, d) damp river DN load increases in cases with relatively large fertilizer inputs (SSP1-2.6, SSP2-4.5), while the opposite holds for

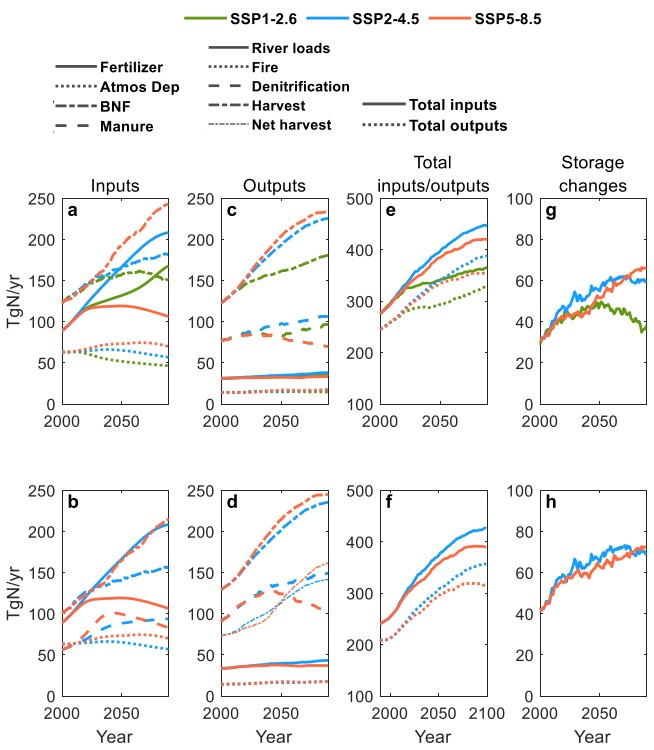

**Fig. 2 | Terrestrial and freshwater N storage and fluxes. a–h** N inputs (**a, b**), outputs (**c, d**), total inputs and outputs (**e, f**), and changes in terrestrial and freshwater N storage (**g, h**) in simulations without manure applications (**a, c, e, g**) and with manure applications (**b, d, f, h**). Total inputs are the sum of fertilizer applications, atmospheric deposition, and biological N fixation (BNF). Total outputs are the sum of river loads to the ocean, net harvest (i.e., N in harvested wood, crops, and grasses after subtracting out internally recycled inputs, i.e., manure applied to croplands and grasslands), soil and freshwater denitrification, and fire emissions. Terrestrial and freshwater storage includes 5 vegetation storage (leaves, fine roots, sapwood, heartwood, and labile storage), 4 organic soil storage (fast and slow litter, slow and passive soil), 2 inorganic soil storage (ammonium and nitrate), and 6 freshwater storage (river DON, ammonium, and nitrate, lake DON, ammonium, and nitrate). All plots show 21-year moving averages from 2000–2099. Source data are provided as a Source Data file.

scenarios with low fertilizer additions (SSP5-8.5), reducing the gaps one would expect based on fertilizer alone.

Fertilizer applications in the scenarios are determined by crop demands for food/feed and 2nd generation biofuel (e.g., energy-efficient, flexible feedstock ethanol and methanol from woody biomass) and agricultural productivity (i.e., the amount of crop production per unit area) (Table 1)[13]. These, in turn, depend on population dynamics, dietary preferences, climate policies, and agricultural advances.

In the fossil fuel-intensive scenario SSP5-8.5, the demand and subsequent production of food and feed crops increase the most among the three scenarios considered herein (Fig. 3 and Supplementary Table S6). This occurs despite low population growth, reflecting the assumed prevalence of diets with high animal and waste shares[13,37] (Table 1). The meat-rich diet also underlies the sharp increase in manure applications which peak in the 2040 s, before decreasing after the 2060s due to limited grassland availability, especially in African and Latin American regions[39–41]. Despite the highest food and feed crop production, projected fertilizer applications in SS5-8.5 increase only slightly (Fig. 1c) due to very low biofuel crop production in the absence of explicit climate mitigation policies (Fig. 3) and rapid improvements in agricultural productivity[13,37] (Table 1). River DN loads are projected to increase by only 7% (without manure applications, Fig. 1a) or 8%

(with manure applications, Fig. 1b) from the present (2000–2020) to the end of the 21st century (2079-2099). This occurs despite the largest increase in atmospheric N deposition and BNF in SSP5-8.5 (Fig. 2a, b).

The positive outcomes in terms of water N pollution projected in SSP5-8.5, however, are likely accompanied by less favorable air pollution outcomes. Relatively high harvest (i.e., N in harvested woods, crops, and grasses) or net harvest (i.e., harvest after subtracting out recycled inputs, i.e., manure applied to croplands and grasslands, Fig. 2c, d), as well as high atmospheric deposition (Fig. 2a, b), in SSP5-8.5 is indicative of a high atmospheric N pollution potential. This is because much of the net harvest is likely to ultimately go to the atmosphere via wood, biofuel, and waste burning or emissions from food, human, and livestock waste. While $N_2$ emissions via this pathway are generally considered benign, and some harvested N is sequestered in durable goods (i.e., home building), a significant fraction would likely be emitted as NOx, $N_2O$, and $NH_3$[39,45].

In SSP2-4.5, dietary changes associated with climate mitigation efforts temper growth in food and feed crop production (Fig. 3) despite higher population growth than in SSP5-8.5[13,36] (Table 1). However, climate mitigation-driven biofuel crop production exceeds that required for food and feed (Fig. 3). Growing food/feed and biofuel demands in SSP2-4.5 ultimately result in the largest increase in crop production among the three scenarios. This combines with less rapid agricultural advances than SSP5-8.5[13,36] (Table 1) to create the largest fertilizer requirement among the three scenarios (Figs. 1c, 3), increasing global river DN loads by 21% (without manure applications, Fig. 1a) or 25% (with manure applications, Fig. 1b).

In the sustainability scenario SSP1-2.6, low population growth, low-meat diets with low food waste, and advances in agricultural systems limit growth in fertilizer demand for food and feed crop production, but these factors are countered by biofuel-driven fertilizer demand associated with ambitious climate mitigation efforts[13,35] (Table 1, Fig. 3). Strong air pollutant controls, high shares of renewables, and rapid energy efficiency improvements decrease atmospheric deposition markedly (Fig. 2a, b)[35,46], but not by enough to avoid moderately increasing N inputs (Fig. 2e) leading to a 14% river DN load increase that lies between the SSP5-8.5 and SSP2-4.5 responses (Fig. 1a). Given the relatively low livestock production in SSP1-2.6 (Supplementary Fig. S5), manure applications would likely enhance this increase by a similarly modest or lesser value than the 1-4% enhancements projected by SSP5-8.5 and SSP2-4.5, yet would not be enough to shift the order imposed by the stark differences in fertilizer applications.

Comparison of the projections with all forcings with those omitting $CO_2$ fertilization (Fig. 1a, b dotted lines) suggests that $CO_2$ fertilization could slow the rate of river DIN load increase, with greater effects in scenarios with higher greenhouse gas emissions (SSP5-8.5) and higher fertilizer applications (SSP2-4.5). That is, river DIN loads during 2079–2099 are 15% and 14% (without manure applications, Fig. 1a) or 21% and 15% (with manure applications, Fig. 1b) lower than those without $CO_2$ fertilization effects in SSP5-8.5 and SSP2-4.5 respectively. This reflects increases in plant uptake and subsequent storage (Fig. 2e–h), consistent with previously described historical simulations[20] (see Discussion). The $CO_2$ fertilization effect is less marked in the low greenhouse gas emission scenario SSP1-2.6.

While the global analysis above highlights the wide range of socioeconomic development and climate policy implications for future water N pollution, a deeper understanding of the interplay between these factors requires a regional perspective. Analysis of 11 aggregate major world regions shows contrasting river N responses across a spectrum from low to high income (i.e., pcGDP[42]) regions (Figs. 4, 5a, b).

In high income (i.e., high pcGDP) regions, including North America (NAM), Pacific OECD (PAO, e.g., Australia), and Western Europe (WEU), river DN loads across the three scenarios are

**Table 1 | Socioeconomic drivers of fertilizer and manure applications, and atmospheric deposition**

| | SSP5-8.5 | SSP2-4.5 | SSP1-2.6 |
|---|---|---|---|
| Socioeconomic drivers of fertilizer applications[13] | | | |
| Food & feed crop demand | Medium | Low | Low |
| Population (billions by 2100) | Low (7.4) | Medium (9.0) | Low (7.0) |
| Dietary preferences | Meat-rich, high waste shares | Dietary changes for climate mitigation | Low meat, low waste shares |
| Biofuel crop demand | Very low | Relatively high | Relatively high |
| Radiative forcing W m$^{-2}$ by 2100 from baseline to climate mitigation scenarios | 8.7 (Baseline) | From 6.5 to 4.3 | From 5.0 to 2.6 |
| Agricultural productivity growth | High | Medium | High |
| Advancements in agricultural technologies, practices, and management | Rapidly increasing crop yields, highly managed agricultural systems, improved feeding efficiencies | Moderate crop yield growth largely achieved by additions of fertilizers, medium feed conversion efficiencies | Rapidly increasing crop yields, rapid diffusion of best practices, high feed conversion efficiencies |
| Agricultural trade | Very high | Medium | Low |
| Socioeconomic drivers of manure applications[40,41] | | | |
| Livestock production | High | Medium | |
| Grassland availability | Medium | Medium | |
| Socioeconomic drivers of atmospheric deposition[46] | | | |
| Air pollution controls | Strong | Medium | Strong |
| Technological innovation, and other relevant socioeconomic conditions | Less polluting fuels and technologies | Moderate technological innovation | Less polluting fuels and technologies, increasing use of public transport, rapidly increasing access to clean energy for cooking in developing countries |
| Air quality cobenefits from climate mitigation | None (no explicit climate policies) | Medium | Low (due to strong pollution controls) |

projected to generally decline or stabilize (-18 to 17% from 2000–2020 to 2079–2099, Figs. 4, and 5a, b). This is the case even under climate mitigation scenarios (SSP1-2.6, SSP2-4.5) featuring significant global DN load increases (Fig. 1a, b). High pcGDP regions show relatively stable or decreasing N inputs (Fig. 5c–g, and Supplementary Fig. S6) and cropland area (Supplementary Figs. S7–S9). In the scenarios, this is achieved through projected improvements in agricultural productivity[23,47], relatively small increases in food and feed production associated with relatively low populations (Supplementary Fig. S10), less meat-intensive diets with low food waste[35], and/or strict and well-enforced air pollution policies[46].

In contrast, low income (i.e., low pcGDP) regions of Sub-Saharan Africa (AFR), South Asia (SAS), and Middle East and North Africa (MEA) exhibit generally increasing river DN loads (9–101%, Figs. 4, 5a, b) and N inputs (Fig. 5c–g, and Supplementary Fig. S6). Fertilizer increases are associated with an expansion of cropland area (Supplementary Figs. S7–S9) and/or intensification of fertilizer applications in less efficient agricultural systems[13,37]. Relatively weak air pollution controls and limited improvements in greenhouse gas emission intensities (i.e., emissions per unit of energy used) are also not sufficient to offset growth in fossil fuel use and other emission drivers[46], thereby increasing or stagnating atmospheric deposition (Fig. 5d, and Supplementary Fig. S6). Projected river DN load increases extend to several moderate income (i.e., moderate pcGDP) regions, most notably Latin America and the Caribbean (LAC) (10–31%) serving as prominent bioenergy producers (see Fig. SI9 of Popp and colleagues[13]) with the large conversion of food/feed cropland for 2nd generation biofuel production (Fig. S7).

As in the global case, additions of manure applications to SSP2-4.5 and SSP5-8.5 did not impact the trend contrasts between low and high pcGDP regions (Figs. 4–5). While manure effects on SSP1-2.6 are uncertain, the secondary importance of these inputs relative to fertilizers (Fig. 1c) limits their impacts and even partly compensating responses would not eliminate equity concerns imposed by fertilizer trends.

The projected contrasts in river DN load trends between high and low pcGDP regions are generally smaller under SSP5-8.5, though they are still apparent. High pcGDP regions in SSP5-8.5 achieve the greatest reductions in river DN loads (-18 to -3%, Figs. 4, and 5a, b), but these are achieved without explicit climate mitigation through biofuel crop production and also benefit from highly globalized agricultural trade that shifts fertilizer burdens associated with food production to low pcGDP regions[13,37]. This differs from SSP1-2.6 and SSP2-4.5, where river DN declines in high pcGDP regions are achieved with relatively low levels of agricultural trade and net exports of agricultural products from high pcGDP regions[13]. However, even while supporting net exports of agricultural products, low pcGDP regions under SSP5-8.5 exhibit more stable river DN loads (10-26%) than under SSP1-2.6 or SSP2-4.5 (9-101%). This is due to very low projected biofuel crop production and/or rapid advances in agricultural technology enabling food and feed crop production without large increases in fertilizer applications[13,37]. Only Africa (AFR), which is projected to experience large increases in atmospheric deposition associated with high consumption of transport fuels[13,37,46], has river DN load increases in SSP5-8.5 comparable to those in SSP1-2.6 and SSP2-4.5.

## Discussion

Our projections emphasize the overarching importance of fertilizer use in determining future global river N pollution despite multiple N inputs, co-occurring climate and $CO_2$ fertilization effects, and the critical role of climate mitigation activities and agricultural advances in future fertilizer use. They also elucidate divergent river N pollution projections and potential inequities between low- and high-income (i.e., pcGDP)[42] regions across the scenarios.

Surprisingly, the scenario without explicit climate mitigation policies (SSP5-8.5) achieves the best outcomes in terms of water N pollution. The favorable outcomes, however, are associated with a higher potential for atmospheric pollution and rely on the assumption of rapid technological advances that are quickly disseminated in a globalized world[13,37]. They also must be weighed against likely severe

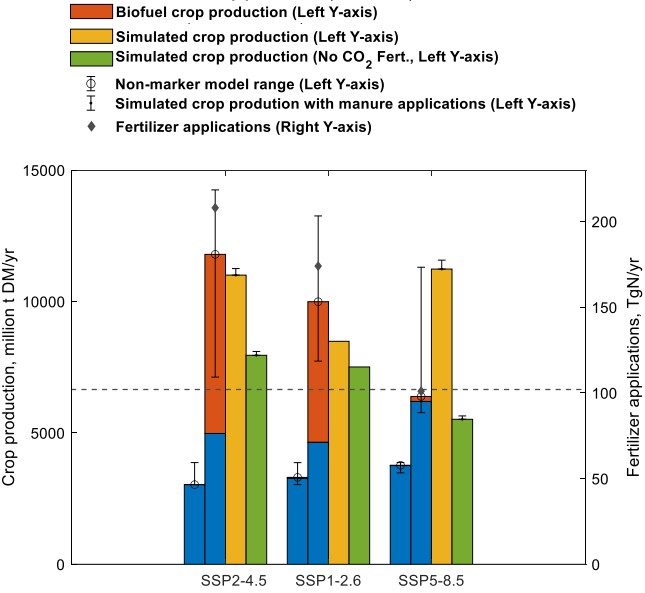

**Fig. 3 | Crop production and fertilizer application.** Left y-axis shows production of food and feed crops in 2010 (left-most blue bars) and in 2100 (second blue bars) and production of 2nd generation biofuel crops (red bars) in 2010 and 2100 from the IAM marker scenarios[13]. Error bars for these reported crop production present ranges of crop production from IAM marker and non-marker scenarios (Supplementary Table S6)[72], reflecting a large uncertainty from the IAM-specific inter-pretation/implementation of SSP narratives. This IAM crop production is compared with the simulated crop production in LM3-TAN marker scenario simulations (yellow bars) and in our uncertainty simulations without $CO_2$ fertilization (green bars) in 2099 (See "Methods"). Error bars for LM3-TAN simulations present crop production in simulations with manure applications. Right y-axis shows fertilizer applications to croplands in 2010 (dashed line) and in 2100 (diamonds)[30]. Fertilizer applications are generally higher in scenarios with higher biofuel crop production, though differences in agricultural productivity between the SSPs can alter the relationship between crop production and fertilizer applications and the inclusion of non-marker IAM scenarios broadens ranges of crop production, indicating an uncertainty on fertilizer applications. Simulated crop production in LM3-TAN is generally consistent with the IAM marker scenarios with the exception of SSP5-8.5, where our projected crop production with $CO_2$ fertilization is slightly above the range of IAM marker and non-marker SSP5-8.5 scenarios. Importantly, impacts of climate change and $CO_2$ fertilization, which would be greatest in SSP5-8.5, were not taken into account for when estimating crop production and associated fertilizer requirements in the IAM marker scenarios[13]. Although such impacts are uncertain, removing them creates SSP5-8.5 crop production in LM3-TAN consistent with the IAM marker SSP5-8.5 crop production. LM3-TAN thus suggests that $CO_2$ fertilization under high emission scenarios like SSP5-8.5 may reduce fertilizer requirements relative to what IAMs suggest. Source data are provided as a Source Data file.

climate change impacts under this highest emission pathway[48]. Under the mitigation pathways SSP1-2.6 and SSP2-4.5, the adoption of more sustainable, low-meat diets with low food waste tempers the growth of fertilizer applications for food and feed. These gains, along with agricultural yield-increasing technologies, efficient livestock nutrition, and waste management[23,47,49], are enough to balance bioenergy-associated burdens in high-income regions. They are, however, not sufficient to avoid large river DN load increases in many low- and moderate-income regions projected to undergo rapid expansion of less efficient agricultural systems.

The reductions in river DIN loads under SSP5-8.5 relative to SSP1-2.6 also rest in part with the projected enhancement of vegetation growth and N storage associated with $CO_2$ fertilization in LM3-TAN simulations. This result aligns with reported damping effects of $CO_2$ fertilization on other inorganic N losses, such as $N_2O$ emissions from soils[50] and rivers[51], implying that models that do not account for $CO_2$

fertilization effects[17,24–26,52] may overestimate future river DIN loads, especially under high emission scenarios. In addition, such projections would under-represent a modest, yet significant projected shift in riverine loads toward dissolved organic phases following $CO_2$ fertilization-induced, enhanced vegetation growth and storage[19–23] (Fig. 1a, b and Fig. 2g, h). This could be critical for water quality since organic N contributes less to acute eutrophication responses in coastal areas than more bioavailable inorganic counterparts[53].

It is notable that the three CMIP6 IAM marker scenarios applied here did not consider $CO_2$ fertilization effects (which would be greatest in SSP5-8.5) when estimating crop production and associated fertilizer requirements[13]. Our results support the potential positive impact of $CO_2$ fertilization on crop production arising from improved N use efficiency (NUE) and crop productivity[19,21,23] (Fig. 3), although there is considerable uncertainty in the effects of elevated $CO_2$ on crop yields[54,55]. Accounting for such effects in the scenarios thus could further reduce N fertilizer requirements and river DIN loads in SSP5-8.5 relative to other scenarios.

Our results show insignificant or modest negative relationships of projected river DN load changes with precipitation changes from 2000–2020 to 2079–2099 across the 11 regions and three scenarios (r = −0.10, p = 0.40) or temperature changes (r = −0.40, p < 0.01, Supplementary Figs. S2–S4). This reflects the predominance of N inputs in determining future river DN loads, and the array of processes controlling the DN loads realized at river mouths (Fig. 2). A systematic exploration of the changing precipitation and warming impacts would require a new experimental design and is left to future work. Nonetheless, the limited relationships found here suggest caution when applying empirical precipitation or temperature relationships to future projections, even when they have proven contemporary skill.

With this in mind, it is useful to compare our process-based projections with those of Sinha and colleagues who enlisted an empirical model based on annual precipitation, extreme springtime precipitation, net anthropogenic N inputs, and land use to project river total N loads[17]. They projected an 8% total N load increase from the continental United States during 2071–2100 relative to 1976–2005 under SSP5-8.5, concluding that impacts of reduced anthropogenic N inputs (yielding a 7% load decrease) will be exceeded by load enhancements due to increasing precipitation. In contrast, we project limited net impacts of precipitation changes and warming in the continental United States for the same period, resulting in a 9% and 6% decrease in river DN loads without and with manure applications. In the case of mitigation scenarios, Sinha and colleagues project a 13% and 1% increase in loads under SSP1-2.6 and SSP2-4.5 respectively, while our results project a 3% increase under SSP1-2.6 and a 3% and 17% increase without and with manure applications under SSP2-4.5. We stress, however, that the limited net effects of changing precipitation and warming over multidecadal and large regional scales do not imply that hydroclimate changes will not impact water quality, as loading extremes and timing in specific regions can be consequential. It has been suggested, for example, that severe hydroclimate extremes expected under high emission scenarios may yield more localized acute impacts on river N loads[18].

Several limitations of our modeling framework should be considered when interpreting these results. First, while CMIP6 forcing datasets provide one of the most comprehensive future global land use, anthropogenic N, and $CO_2$ inputs to land biosphere models, they do not provide recycled N inputs. This limitation has been partly resolved by using the future manure applications constructed based on the CMIP6 forcing datasets[39–41]. Although the current scientific literature does not provide documented future manure applications for SSP1-2.6, the results based on the available manure projections suggests they are secondary to fertilizers both in the overall magnitude of the inputs and the contrasts between scenarios. It would be, however, best to have an explicit manure estimate for SSP1-2.6, and additional

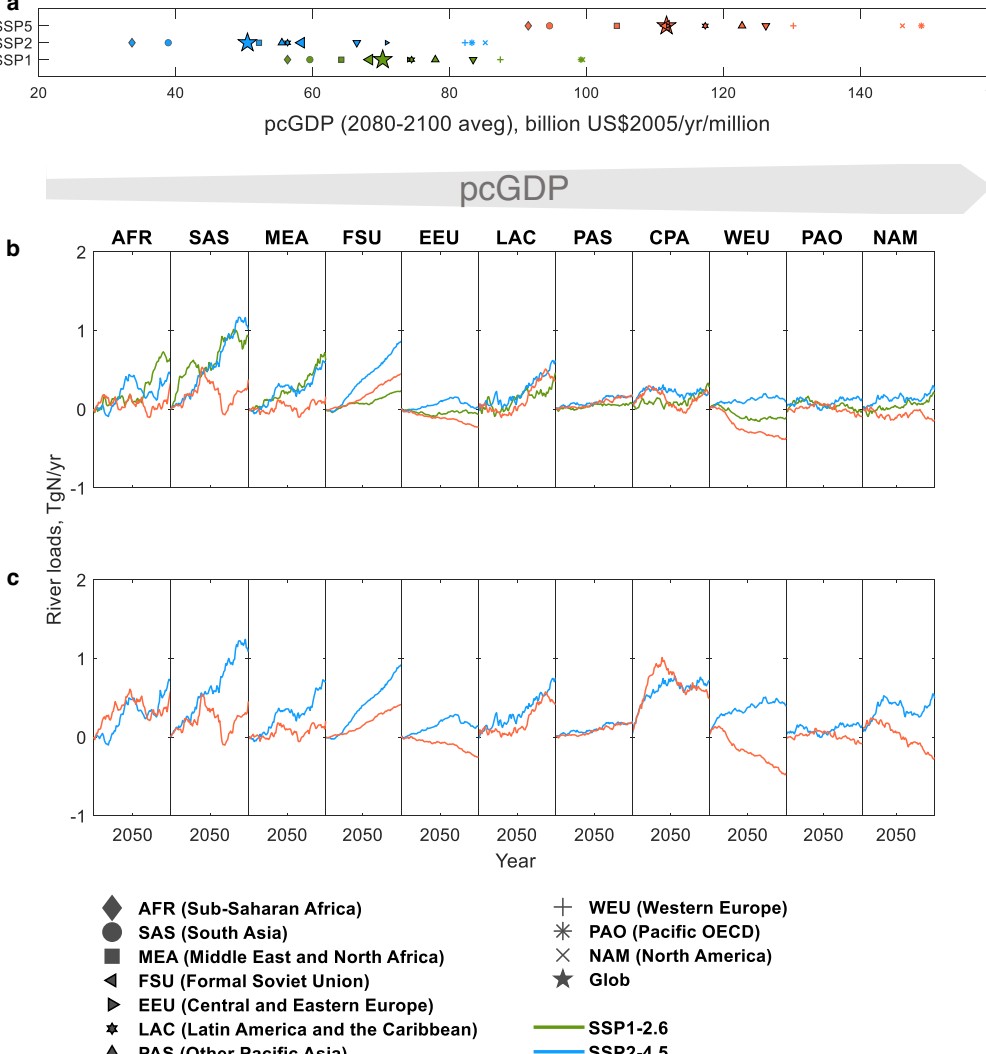

**Fig. 4 | Regional variation in river DN loads. a** 2080–2100 average per-capita gross domestic product (pcGDP) for 11 aggregate regions (Supplementary Fig. S12, indicated by shapes) for each of the three scenarios (colors). **b, c** River DN load anomalies from 11 aggregate regions to the coastal ocean without manure applications (**b**) and with manure applications (**c**). Panels are arranged by increasing pcGDP from left to right. All plots show 21-year moving averages from 2000–2099. Notice the prevalence of increasing river DN loads in the left panels, and the prevalence of decreasing or stabilizing river DN loads in the right panels. Source data are provided as a Source Data file.

assessment of manure dynamics is needed. Our approach also does not explicitly account for sewage, but its importance will be likely much smaller than manure applications[24]. Ultimately, these uncertainties will be better explored when N-explicit scenarios, based on new N-focused narratives[56], become available to force land biosphere models.

Second, our model does not resolve the potential impact of phosphorous limitation on $CO_2$ fertilization effects. This may be particularly significant in low phosphorus systems like tropical intact forests[57]. Elevated atmospheric $CO_2$, however, has been identified as a main driver of observed increases in aboveground biomass or C uptake in many intact forests[22,58,59] and our global response to $CO_2$ fertilization is quantitatively consistent with a recent empirical estimate[20]. While these limitations are being targeted in ongoing work, the considerable fidelity of current simulations with observed river N concentrations/loads and published terrestrial and/or freshwater N and C budgets (Supplementary Fig. S1, Supplementary Table S4, and Supplementary Note S1, Lee and colleagues[18,20]) supports the utility of current results.

While uncertainties in N inputs across future scenarios and the earth system response to these inputs remain, our results highlight the need for careful consideration of unintended, adverse effects of bioenergy-driven climate mitigation efforts on water quality and the inequity of outcomes along a socioeconomic spectrum. Projections in high-income regions provide cause for optimism for meeting climate mitigation targets and decreasing or stabilizing water N loads relative to current levels, but a commitment to rapid transfer of agricultural advances to low-income regions and bioenergy production options without raising fertilizer applications[60] may be required to meet these goals across the economic spectrum.

## Methods

### Terrestrial-freshwater ecosystem model LM3-TAN

The Geophysical Fluid Dynamics Laboratory (GFDL) Land Model LM3-Terrestrial and Aquatic Nitrogen (TAN)[20,29] simulates coupled water, C, and N cycles within a vegetation-soil-river-lake system[61–63]. LM3-TAN simulates the distribution of five plant functional types (C3 and C4 grasses, temperate deciduous, tropical, and cold evergreen

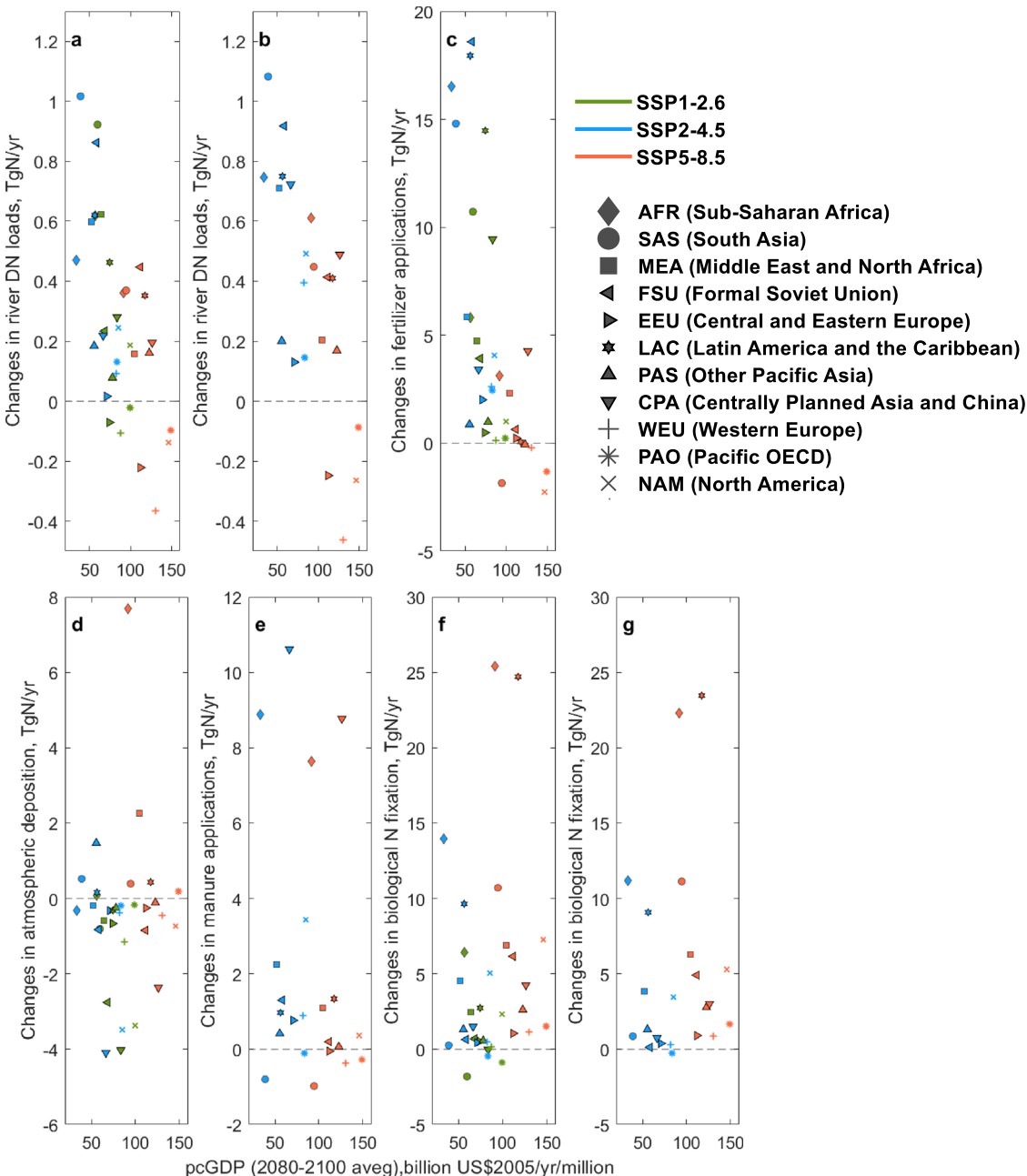

**Fig. 5 | Changes in river DN loads and N inputs. a–g**, 2080–2099 average pcGDP vs. changes in river DN loads without manure applications (**a**) and with manure applications (**b**), fertilizer applications (**c**), atmospheric deposition (**d**), manure applications (**e**), biological N fixation without manure applications (**f**) and with manure applications (**g**) between 2000–2020 and 2079–2099. Source data are provided as a Source Data file.

trees) based on prevailing climate conditions and C–N storage in vegetation. Scenarios of land-use states and transitions are used to simulate four land-use types: primary lands (lands effectively undisturbed by human activities), secondary lands (abandoned agricultural land or regrowing forest after logging), croplands, and pastures. Unlike most empirical watershed models that require stationarity assumptions[26], LM3-TAN captures key terrestrial dynamics that affect the state of vegetation and soil C-N storage, such as vegetation growth, leaf fall, natural and fire-induced mortality, deforestation for agriculture, wood harvesting, reforestation after harvesting, crop harvesting, grazing, and various soil microbial processes, which are described in detail elsewhere[29,61,63]. LM3-TAN is thus well suited to simulate the delivery of N from terrestrial systems to rivers and coastal waters.

Terrestrial runoff of three N species (nitrate, ammonium, and DON) to river systems is subject to retention within rivers and lakes or transformed during transport to the coastal ocean[20,29]. Each model grid cell contains one river reach and/or one lake, in which freshwater microbial processes (i.e., mineralization, nitrification, and denitrification) are simulated. First-order kinetics are used to describe the freshwater microbial processes. Reaction rate coefficients of freshwater mineralization and nitrification are prescribed. Reaction rate coefficients of freshwater denitrification are estimated by a reported nonlinear regression function based on Lotic Intersite Nitrogen experiment reach-scale measurements[64–66]. An Arrhenius-based relationship is used to adjust the rate coefficients for temperature effects, approximately doubling the rates for a temperature increase of 10 °C. Water containing N in each river reach or lake flows to another river

reach in the downstream grid cell following a network that ultimately discharges to the ocean. Freshwater physics, hydrology, and hydrography are described in detail elsewhere[62].

N inputs to LM3-TAN include simulated BNF, fertilizer applications, and atmospheric deposition. N outputs from LM3-TAN include river loads to the ocean, emissions to the atmosphere, and net harvest (N in harvested wood, crops, and grasses after subtracting out internally recycled inputs, e.g., manure applied to croplands and sewage). See references[20,29] for the detailed model descriptions and schematic diagram.

LM3-TAN was globally implemented at 1 degree spatial and 30 minute temporal resolution to simulate the past two and half centuries of terrestrial and freshwater C and N storage and fluxes to the ocean and atmosphere, which were found to be consistent with published syntheses from 28 different studies, when comparable categorization, definitions, and assumptions were applied[18,20]. With updated forcings and minor calibration refinements to make future projections herein (see below), we show that simulated river water discharge, nitrate, DIN, and DON concentrations and loads agree well with measurement-based estimates across 49, 36, and 21 major rivers, which are distributed broadly over the globe and influenced by various climates, biomes, and human activities (Supplementary Note S1, Supplementary Tables S1–S4, and supplementary Fig. S1). LM3-TAN results of global soil N storage and global river DIN and DON loads are also consistent with published estimates (Supplementary Note S1, Supplementary Table S4). Although observations of global and large regional land N storage and flux changes over long time periods are not available for model validation, LM3-TAN results of global terrestrial C storage and flux changes (which are closely linked with those of N) on centurial to multi-year time scales are well within ranges from large-scale constraints and atmospheric studies (Supplementary Note S1). LM3-TAN's capacity to simulate N accumulation within or release from vegetation, soil, and freshwater storage, in response to changes in atmospheric $CO_2$, climate, anthropogenic N inputs, and land use and land cover, supports projections for longer time horizons, during which unprecedented socioeconomic and climate changes will push river basins well beyond the scope of prior observations[27,28].

All the processes in LM3-TAN are updated every 30 minutes. This means that the model outputs (including river water and N throughout the river network) are given at the 30 minute time step. The annual results for water discharge, N loads and concentrations are the averages of the 30 minute results with applicable unit conversions. These high temporal resolution, continuous outputs do not require discharge-weighted calculations or any other additional result manipulations.

## LM3-TAN marker simulations under socioeconomic climate scenarios

Following ~12,600 years of spin-up, LM3-TAN was globally implemented at 1 degree resolution to simulate global terrestrial and freshwater N storage and fluxes to the ocean and atmosphere from 1700 to 2014 under pre-industrial and historical forcings. Projections to 2099 were then conducted under three Coupled Model Intercomparison Project, Phase 6 (CMIP6) Integrated Assessment Model (IAM) marker scenarios: Shared Socioeconomic Pathway (SSP)1-2.6[35], SSP2-4.5[36], and SSP5-8.5[37].

In a Scenario Matrix Architecture[67], SSPx-y scenarios combine possible future socioeconomic developments described in SSPx[68] with shared climate policy assumptions (SPAs)[69] to reach Representative Concentration Pathway (RCP)-specific radiative forcing levels y (W m$^{-2}$)[38]. SSP1 (Sustainability – Taking the Green Road) features sustainable development reducing environmental degradation and inequality costs (Table 1). SSP2 (Middle of the Road) features the world with social, economic, and technological trends

similar to historical patterns: progress toward sustainability characterizing SSP1 is tempered by more rapid population growth, lower stringent pollution controls, and less rapid technological advances. Finally, SSP5 (Fossil-fueled Development – Taking the Highway) features rapid and fossil fuel-driven development with high economic growth. SSP1, SSP2, and SSP5 without SPAs (i.e., SSP baselines) reach radiative forcing levels 5 W m$^{-2}$ (5.0–5.8 W m$^{-2}$), 6.5 W m$^{-2}$ (6.5–7.3 W m$^{-2}$), and 8.7 W m$^{-2}$ respectively[70]. Notably, the rapid and fossil fuel-driven development of SSP5-8.5 is built around a SSP5 baseline scenario without explicit climate change mitigation policies and without considering impacts of climate change and $CO_2$ fertilization[37], leading to the upper end of radiative forcing levels in the scenario literature[38,71].

Climate forcing for the pre-industrial, historical, and future simulations was taken from GFDL's CMIP5 Earth System Model ESM2Mb[33,34]. The climate forcing includes precipitation, specific humidity, air temperature, surface pressure, wind speed, and short- and long-wave downward radiation for each of the three radiative forcing trajectories in the marker scenarios. CMIP6 forcing datasets were used for atmospheric $CO_2$[31,32], land-use states and transitions[30], fertilizer applications[30], and atmospheric deposition[31]. Manure applications constructed by using the CMIP6 forcing datasets were taken from the dynamical agricultural module (CAMEO, Calculation of AMmonia Emissions in ORCHIDEE) within the land surface model ORCHIDEE[39–41].

For land use and fertilizer calculation, 12 land-use types reported in the Land Use Harmonization (LUH2)[30] were grouped into 4 types in LM3-TAN: 1) primary land in LM3-TAN is the sum of forested primary land and non-forested primary land in LUH2, 2) secondary land in LM3-TAN is the sum of potentially forested secondary land, potentially non-forested secondary land, and urban land, 3) cropland in LM3-TAN is the sum of C3 annual cropland, C3 perennial cropland, C4 annual cropland, C4 perennial cropland, and C3 N-fixing cropland, and 4) pasture in LM3-TAN is the sum of managed pasture and rangeland. The sum of fertilizers allocated to the 5 croplands in LUH2 was applied to the cropland in LM3-TAN. Atmospheric deposition was applied at equal rates to all land-use types.

Manure applications are available for the historical period 2002-2014 and for the future period 2015–2100 for the SSP2-4.5, SSP4-3.4, and SSP5-8.5 scenarios, with SSP4-3.4 and SSP5-8.5 respectively representing the lowest and highest pathways of livestock production and manure applications among the CMIP6 IAM scenarios[40,41,72]. The process-based model CAMEO, which was developed and validated by Beaudor and colleagues[39], was used to estimate manure dynamics. In order to project future manure production and applications, CAMEO was run with updated inputs. The updated inputs are climate forcings taken from the Earth System Model IPSL-CM6A-LR[73], the same CMIP6 forcing datasets as used in this study (i.e., land use[30], fertilizer applications[30], atmospheric deposition[31], and livestock production[72]), and reconstructed gridded future livestock densities. Future livestock production provided for large regions[72] was downscaled to the grid-cell level by assuming that grass-feed livestock demand was met locally, based on future trends in grassland areas, above-ground net primary productivity of the grassland, and a Grazing Intensity parameter introduced in Beaudor and colleagues[39].

Manure from managed lands (after accounting for removals from animal housing, storage, etc) was applied to cropland and pasture in LM3-TAN. Manure from grasslands was applied only to pasture in LM3-TAN. For the 1861-2001, manure applications to pasture and cropland were estimated from grazing and crop production in LM3-TAN. Fractions of grazing and crop production that become manure applications and fractions of partitioning between N species were based on the reported historical manure applications (Note S2). This provides a smooth transition between past, current, and future manure

applications for our experiments that include manure (Fig. S11). We emphasize that this admittedly uncertain approach has little impact on the differences in projected values of primary interest herein.

## Climate and $CO_2$ uncertainty simulations

Acknowledging the uncertainties arising from future climate and atmospheric $CO_2$ changes, we tested the uncertainty of river N loads to the following alternative conditions:

1. Precipitation in SSP5-8.5 was proportionally increased to yield a 7.6% (8.2% over land) increase by 2080-2099 relative to 1986-2005. This simulation run from the year 2020 evaluates an impact of the upper end of precipitation increases from multi model simulations (IPCC, 2007)[43].

2. Air temperature in SSP5-8.5 was proportionally increased to yield a 5.3 °C (6.0 °C over land) increase by 2080-2099 relative to 1986-2005. This simulation run from the year 2020 evaluates an impact of the upper end of temperature increases from multi model simulations (IPCC, 2014)[44].

3. Atmospheric $CO_2$ concentration in all three scenarios remained as the level of the year 2014 until 2099. These simulations without future $CO_2$ fertilization are aimed at evaluating impacts of future $CO_2$ fertilization.

## Aggregate major world regions

The results are presented and discussed for 11 aggregate major world regions defined at https://iiasa.ac.at/web/home/research/Flagship-Projects/Global-Energy-Assessment/GEA_Annex_II.pdf: (1) North America (NAM), (2) Pacific OECD (PAO), (3) Western Europe (WEU), (4) Centrally Planned Asia and China (CPA), (5) Other Pacific Asia (PAS), (6) Latin America and the Caribbean (LAC), (7) Central and Eastern Europe (EEU), (8) Formal Soviet Union (FSU), (9) Middle East and North Africa (MEA), (10) South Asia (SAS), and (11) Sub-Saharan Africa (AFR). As river basins are not perfect subsets of the 11 country groups, small deviations between the 11 regions and our basin-scale groups mapped to the 11 regions exist (Supplementary Fig. S12).

## Data availability

Source data used to produce all of the figures in the main text and supplementary information are provided in the Source Data file with this paper. Table S7 provides a description of all used inputs and links to all of the datasets that are publically available. All the other publically unavailable inputs are available at https://zenodo.org/records/10962725[74], except the one (high-frequency atmospheric forcing data) which is very large (totaling 899 GB), greatly exceeding the ~50 GB limits of commercial data storage services like Zenodo. This dataset is available from the corresponding author upon request. Source data are provided in this paper.

## Code availability

The model codes are available at https://doi.org/10.5281/zenodo.11200697[75].

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

## Acknowledgements

We thank J. Dunne and A. Ross from the National Oceanic and Atmospheric Administration (NOAA)/Geophysical Fluid Dynamic Laboratory (GFDL) for their incisive comments on the manuscript. Award NA23OAR4320198 from the NOAA, U.S. Department of Commerce (M. Lee). The statements, findings, conclusions, and recommendations are those of the authors and do not necessarily reflect the views of NOAA, or the U.S. Department of Commerce. M. Beaudor and N. Vuichard acknowledge the financial support received via the ESM2025 project from H2020 European Institute of Innovation and Technology (grant no. 101003536). In addition, M. Beaudor acknowledges the work environment and resources provided at LSCE including the GENCI supercomputer system (Joliot-Curie supercomputer).

## Author contributions

M.L., C.A.S., E.S. and S.M. contributed to developing the model and its global implementation. M.L. performed the model simulations and analyses. M.B. and N.V. created the manure application datasets. All authors analyzed and discussed the results. M.L. and C.A.S. wrote major portions of the manuscript with substantial inputs from E.S and S.M.

## Competing interests

The authors declare no competing interests.
