## [Peer Review File · Nature Communications]

REVIEWER COMMENTS

Reviewer #1 (Remarks to the Author):

Lee et al review

This article is an important contribution to the literature, presenting new estimates of river nitrogen loads across the 21st century under different climate and socioeconomic scenarios. Using a finely detailed, well-tested terrestrial-freshwater ecosystem model, the authors are able to generate nuanced, regionally-specific estimates of river nitrogen loads that reveal the consequences of different land-use management decisions. Before recommending this article for publication, there are four general comments and several more specific ones that the authors are requested to address:

1. The authors only consider synthetic fertilizer and atmospheric deposition as anthropogenic N inputs. They do not consider manure or biological N fixation from cultivated legumes. This is a significant gap in their analysis, and while they do acknowledge it and address it via uncertainty ranges, this should at the very least be made more explicit in the text and ideally should be somehow integrated into their analysis, especially of scenario SSP 5-8.5. This is because, while river DN loads from synthetic fertilizer may be lowest in SSP5-8.5, river DN loads from manure may be highest in SSP 5-8.5 due to high levels of livestock production. Consequently, their results and scenario ranking may be very different when considering all forms of N inputs.
2. Linked to this, it is important to acknowledge that all forms of N pollution are linked via the N cascade (Galloway et al. 2003) and so ideally total N losses should be considered here, including NH₃, N₂O and NO_x. As the authors note, while SSP5-8.5 has the lowest river DN load by the end of the 21st century, it has the highest increase in atmospheric N pollution. So the exclusive focus on river N is only part of the story. I don't mean for this to suggest that the authors should model the entire N cycle, but they need to make this point much more explicitly in their paper.
3. Thirdly, the authors should consider using N-explicit scenarios that were recently developed based on the SSPs and RCPs. They explicitly add on N policy ambition levels which would make them a more relevant family of scenarios for this paper. See Kanter et al. 2020 in *Global Environmental Change* ("A framework for nitrogen futures in the shared socioeconomic pathways").
4. Finally, it might be worth having a native English speaker review the paper to improve the flow of some sentences and increase the clarity of some messages.

Specific comments:

p.1 lines 27-30: Highlight the other aspects of N pollution here, including multiple compounds and environmental human health impacts and its unique chemistry (i.e. the N cascade).

p.2 lines 42-44: This is a very general statement here. Make it clear that these terrestrial and freshwater ecosystem processes including other N losses to air and plant uptake.

p.2 line 46: The low fertilizer additions aspect of SSP5-8.5 will be confusing to some readers who see it as

a “worst case” scenario so make sure to explain this properly. This may also be the place to include mention of higher livestock production and thus manure.

Reviewer #2 (Remarks to the Author):

Uneven pressure of global climate mitigation on regional water quality in the 21th century by Lee et al.

The authors are exploring three scenarios (SSP1, SSP2 and SSP5) and the impact on the dissolved nitrogen export to the ocean. They see that SSP5 leads to the lowest nitrogen export, and SSP2 the highest. Also they suggest that the high income countries (low GDP per capita) can meet their water N level and climate mitigation targets, but the low income countries do not meet both thresholds.

General remarks

I am thinking about the title. The authors conclude here that (line 6, 1-3) there is an insignificant relationship between precipitation and temperature with dissolved nitrogen export. So what do the authors mean by “global climate mitigation”?

It is very hard to follow all the fluxes. I think I would like to have a kind of scheme with the following items for the three scenarios and for current situation and end of the century situation: total N input (and the different sources fertilizer, manure, N fixation and deposition), total N production (harvest and so on), N storage in soils, N load to the river, N export to the coast.

The model LM3-TAN is a complex model. The land processes are described a lot, but the river processes are still not clear to me. They need a good description. The validation of N in rivers is a major concern (see comments of Line 2, 8).

One of the outcomes of this study is that SSP5 is the best scenario regarding water quality. This is an outcome that surprised me. Therefore I dived into the material (Hurt et al, 2020 and Popp et al. 2017) to see how the scenarios were constructed. Fertilizer is the primary determinant of future nitrogen load. These are coming from Hurt et al. 2020. The used fertilizer is originated from three different models (SSP1: IMAGE, SSP2: MESSAGE and SSP5: MAgPIE. Background on these models can be found in Popp et al. 2017. There the following line is found: “Interestingly, the ranking within models is consistent, but the absolute values are strikingly different and mixing and matching models can be misleading.” My question is here: is this comparison of three scenarios misleading, because it is based on three different models? Is it not better to take the three scenarios from one model?

The second point that surprised me is the low fertilizer use in SSP5. The reason for the low fertilizer use (of what I understand of this) is the fact that there is a meat-rich diet, which means that there is a lot of manure available to be used on agricultural land. I don’t understand how this model is accounting for manure (at least I could not extract this from this article), but taking manure into account on a proper way could perhaps lead to a different conclusion. So this is also a major concern.

Remarks

Line x,y means page x, line y

Line 1, 28: “fertilizer runoff”?

Line 1, 28 – 30: Very blunt to assign all impacts to fertilizer runoff. Change.

Lines 1, 43 – 44: Here it is referred to 18 and 20. Both publications have considered the integrated

effects. For 18 it is clear, because it is part of an integrate assessment model “IMAGE”. Number 20 is a set of regression equations. Basic data to feed this model are also from the IMAGE model.

Line 1, 45-46: Why is a projection to the mid -21 century giving a limiting perspective?

Line 2, 2: “most prior projections” and ending with two references which are an example. But why “most”? Give some examples of both cases.

Line 2, 8: LM3-TAN is referenced by 14 and 23. 23 is only for part of the Mississippi but this is a global application. End of this paragraph, the validation is mentioned of this model. Figure S1 gives a very clear perspective. The discharge of DIN and DON are simulated too high. With a higher simulated discharge you would expect lower concentrations to correct this. But no, the model overestimates the concentrations as well. This means that this model has a problem and should be improved. Also the log transformation in the concentrations is not helping and this comparison should be done without log transformation. More elaboration is needed why there is in DIN discharge a location with almost two orders difference. I also looked at the publication number 14. Also here the same problems occur. The discharge looks better in pub 14, but the concentrations are also here to high. What is the difference between the model used in pub 14 and the current model? Where is that described? Table S1 is, in another form (Table S2), also presented in that publication 14. A combination of 14 and table S1 should result in a new table S1 with load, discharge, concentration for DIN and DON with observations and simulations. Unclear for both, is how the simulated values are obtained. How many observations are the base of these observations? Why just an average (and I don’t know which average that is) of the simulated values? I think this is a very minimal validation of this model. This dynamic model wants to capture future projections, but is poorly validated on just one time point. A dynamic model should be validated with long-term observations.

Line 3, 9 In Figs S3-S5 the temperature is 5.3. Here 6.0. Which one is correct?

Line 7, 5: Can you be clear what water N level you mean and what value is used?

Line 9, 30: “20% decrease of NUE”. How is that done? From the first scenario year 20%? The same question for precipitation (line 9, 36) and air temperature (line 9, 39). Which year is the first scenario year in this study?

Line 17, : Figure 3: The SSPs are developed as storylines. Population and GDP were the first parameters that were available. From this starting point all activities are calculated. Some of the parameters in the SSPs are determined based on GDP development. My question is whether fertilizer and atmospheric deposition is created based on direct or indirect GDP development or not? The answer determines whether this figure should be include or not.

Supporting information

Table 1: last line in caption. “Log 10 of discharges” Which unit is used here?

Figure S2a: Changes in soil and freshwater storage. I don’t understand what is included in this. Can you make this more clear (without referring to Lee et al. (2021))

Figure S2a: Is this really 20 year moving average? Looks to me that it is not.

Figure S3 and S4: Explain the regions again.

Reviewer #1 (Remarks to the Author):

Lee et al review

This article is an important contribution to the literature, presenting new estimates of river nitrogen loads across the 21st century under different climate and socioeconomic scenarios. Using a finely detailed, well-tested terrestrial-freshwater ecosystem model, the authors are able to generate nuanced, regionally-specific estimates of river nitrogen loads that reveal the consequences of different land-use management decisions. Before recommending this article for publication, there are four general comments and several more specific ones that the authors are requested to address:

We'd like to thank the reviewer for the positive comments and encouragement. We have done our best to thoroughly address each of the reviewer's comments and feel that they have improved the manuscript.

In this response, the line numbers correspond to those in the original manuscript before making any modifications. Unless noted, table and figure numbers in our responses correspond to the revised manuscript.

1. The authors only consider synthetic fertilizer and atmospheric deposition as anthropogenic N inputs. They do not consider manure or biological N fixation from cultivated legumes. This is a significant gap in their analysis, and while they do acknowledge it and address it via uncertainty ranges, this should at the very least be made more explicit in the text and ideally should be somehow integrated into their analysis, especially of scenario SSP 5-8.5. This is because, while river DN loads from synthetic fertilizer may be lowest in SSP5-8.5, river DN loads from manure may be highest in SSP 5-8.5 due to high levels of livestock production. Consequently, their results and scenario ranking may be very different when considering all forms of N inputs.

With regard to biological N fixation (BNF), we apologize for the misunderstanding: LM3-TAN simulates BNF in all land use types, including BNF in croplands which, as the reviewer points out, is also an anthropogenic input (Lee et al., 2019). To clarify this, we have added a new figure (Fig. 2) and table (Table S5), including all input/output N fluxes and storage changes, and described these in the Results. For details, please refer to our combined response to the reviewer's comment **1** and **2** under the reviewer's comment **2**.

We acknowledge the primary concern of both reviewers regarding uncertainties associated with manure applications. We have identified nascent future manure application projections constructed by using the CMIP6 forcing datasets (Beaudor et al., 2023a, b). Since we have also

used the same CMIP6 forcing datasets for our projections, the manure applications can be and were integrated with our previous analysis.

The future manure applications for the period 2015-2100 are available for SSP5-8.5 and SSP2-4.5. To explicitly estimate the effect of manure applications on future river DN loads, we have run two additional simulations of SSP2-4.5 and SSP5-8.5 with manure applications, in addition to the previous three simulations of SSP1-2.6, SSP2-4.5, and SSP5-8.5 without manure applications. In the runs including manure, historical manure applications available for the period 2002-2014 have been used for the 2002-2014 simulation. For 1861-2001, we have used LM3-TAN to simulate manure applications based on dynamically simulated harvested crops and grazing (details are provided below). The future manure applications have been described in Fig. 2, Fig. 5, and Fig. S5. The manure applications for the entire period 1861-2099 are described in Fig. S10.

The inclusion of the manure applications has increased global river DN loads only modestly and reinforced our primary findings, including that river DN loads are lowest globally and in high-income regions under SSP5-8.5. For SSP5-8.5, global river DN loads are projected to increase by 7% (without manure applications, Fig. 1a) or 8% (with manure applications, Fig. 1b) from the present (2000-2020) to the end of the 21st century (2079-2099). For SSP2-4.5, global river DN loads are projected to increase by 21% (without manure applications, Fig. 1a) or 25% (with manure applications, Fig. 1b).

The results of SSP1, SSP2-4.5, and SSP5-8.5 without manure applications and SSP2-4.5 and SSP5-8.5 with manure applications have been described throughout the manuscript. For most relevant results, please refer to our combined response to the reviewer's comment *1 and 2* under the reviewer's comment *2*.

A description of the manure applications has been added in Methods on page 8, line 35:

Manure applications constructed by using the CMIP6 forcing datasets were taken from the dynamical agricultural module (CAMEO, Calculation of AMmonia Emissions in ORCHIDEE) within the land surface model ORCHIDEE^{31,32}.

And on page 8, line 46:

Manure applications are available for the historical period 2002-2014 and for the future period 2015-2100 for the SSP5-8.5 and SSP2-4.5 scenarios^{31,32}. Manure from managed lands (after accounting for removals from animal housing, storage, etc) was applied to cropland and pasture in LM3-TAN. Manure from grasslands was applied only to pasture in LM3-TAN. For the 1861-2001, manure applications to pasture and cropland were

estimated from grazing and crop production in LM3-TAN assuming fractions of grazing and crop production as becoming manure applications and fractions of partitioning between N species based on the reported historical manure applications (Note S2). This provides a smooth transition between past, current, and future manure applications for our experiments that include manure (Fig. S10). We emphasize that this admittedly uncertain approach has little impact on the differences in projected values of primary interest herein.

The following references are newly added:

Beaudor, M. et al. Global agricultural ammonia emissions simulated with the ORCHIDEE land surface model. *Geosci. Model Dev.* 16, 1053–1081 (2023a).

Beaudor, M., Vuichard, N., Lathière, J. & Hauglustaine, D. Global ammonia emissions from CAMEO throughout the century for 3 scenarios (2000-2100), <https://zenodo.org/records/10100435> (2023b).

2. Linked to this, it is important to acknowledge that all forms of N pollution are linked via the N cascade (Galloway et al. 2003) and so ideally total N losses should be considered here, including NH₃, N₂O and NO_x. As the authors note, while SSP5-8.5 has the lowest river DN load by the end of the 21st century, it has the highest increase in atmospheric N pollution. So the exclusive focus on river N is only part of the story. I don't mean for this to suggest that the authors should model the entire N cycle, but they need to make this point much more explicitly in their paper.

We agree with the reviewer that the negative aspects of SSP5-8.5 in terms of atmospheric N pollution should be noted to provide a balanced perspective of the tradeoffs between the scenarios. We have added a new figure (Fig. 2) and table (Table S5) to explicitly contrast river N loads to the ocean vs. the other N fates. We have also included all N input/output and storage change fluxes in response to the reviewer's previous comments *I*. In addition, we have added a paragraph to the Results discussing the river/atmospheric pollution tradeoffs between the scenarios.

These, as well as the results associated with the manure applications (in response to the reviewer's previous comments *I*), have been described in Results as:

Global river dissolved N (DN) loads to the coastal ocean (hereafter “loads”) are projected to rise in the 21st century under all three scenarios (Fig. 1a, b). Increasing trends in river DN loads are aligned with increasing trends in fertilizer applications (Fig. 1c), rather than

changes in manure applications, atmospheric deposition, and BNF (Fig. 2a, b), or precipitation and temperature (Figs. S2-S4). That is, both river DN loads and fertilizer applications are highest in SSP2-4.5, intermediate in SSP1-2.6, and lowest in SSP5-8.5, while atmospheric deposition, BNF, precipitation, and temperature are highest in the highest emission scenario SSP-8.5, followed by SSP2-4.5 and SSP1-2.6. Manure applications in SSP5-8.5 are higher than those in SSP2-4.5 until the 2060s, but the difference is small relative to fertilizer applications and manure applications in SSP8.5 become lower than SSP2-4.5 by the end of the 21st century. The alignment of trends in river DN loads with those in fertilizer applications remains when uncertain CO₂ fertilization effects (i.e., enhanced vegetation growth due to elevated CO₂) are omitted from LM3-TAN (dotted lines in Fig. 1a, b, see Methods) or when reported extreme limits of climate change uncertainty for SSP5-8.5^{42,43} are considered (+7.6% precipitation or +5.3°C temperature by 2080-2099 relative to 1986-2005, Figs. S2-S4).

While river DN load trends are aligned with fertilizer application trends, the projected magnitude and range of loads across the scenarios (34-43 TgN yr⁻¹ by the end of the 21st century (2079-2099)) is far smaller than those of fertilizer applications (107-208 TgN yr⁻¹, Fig. 1, Table S5). Other inputs and processes in terrestrial and freshwater ecosystems clearly modulate the imprint of fertilizer applications on river DN loads. Relatively low inputs of BNF and atmospheric deposition (Fig. 2a, b) and high denitrification rates (Fig. 2c, d) damp river DN load increases in cases with relatively large fertilizer inputs (SSP1-2.6, SSP2-4.5), while the opposite holds for scenarios with low fertilizer additions (SSP5-8.5), reducing the gaps one would expect based on fertilizer alone.

Fertilizer applications in the scenarios are determined by crop demands for food/feed and 2nd generation biofuel (e.g., energy-efficient, flexible feedstock ethanol and methanol from woody biomass) and agricultural productivity (i.e., the amount of crop production per unit area) (Table 1)¹³. These, in turn, depend on population dynamics, dietary preferences, climate policies, and agricultural advances. Manure applications in the scenarios are primarily determined by livestock production and limited by grassland areas in some regions^{31,32}.

In the fossil fuel-intensive scenario SSP5-8.5, the demand and subsequent production of food and feed crops increase the most among the three scenarios considered herein (Fig. 3, Table S6). This occurs despite low population growth, reflecting the assumed prevalence of diets with high animal and waste shares^{13,39} (Table 1). The meat-rich diet also underlies the sharp increase in manure applications which peak in the 2040s, before decreasing after the 2060s due to limited grassland availability, especially in African and Latin American regions^{31,32}. Despite the highest food and feed crop production, projected fertilizer applications in SS5-8.5 increase only slightly (Fig. 1c) due to very low biofuel crop

production in the absence of explicit climate mitigation policies (Fig. 3) and rapid improvements in agricultural productivity^{13,39} (Table 1). River DN loads are projected to increase by only 7% (without manure applications, Fig. 1a) or 8% (with manure applications, Fig. 1b) from the present (2000-2020) to the end of the 21st century (2079-2099). This occurs despite the largest increase in atmospheric N deposition and BNF in SSP5-8.5 (Fig. 2a, b).

The positive outcomes in terms of water N pollution projected in SSP5-8.5, however, are likely accompanied by less favorable air pollution outcomes. Relatively high harvest (i.e., N in harvested woods, crops, and grasses) or net harvest (i.e., harvest after subtracting out recycled inputs, i.e., manure applied to croplands and grasslands, Fig. 2c, d), as well as high atmospheric deposition (Fig. 2a, b), in SSP5-8.5 is indicative of a high atmospheric N pollution potential. This is because much of the net harvest is likely to ultimately go to the atmosphere via wood, biofuel, and waste burning or emissions from food, human, and livestock waste. While N₂ emissions via this pathway are generally considered benign, and some harvested N is sequestered in durable goods (i.e., home building), a significant fraction would likely be emitted as NO_x, N₂O, and NH₃^{31,44}.

3. Thirdly, the authors should consider using N-explicit scenarios that were recently developed based on the SSPs and RCPs. They explicitly add on N policy ambition levels which would make them a more relevant family of scenarios for this paper. See Kanter et al. 2020 in Global Environmental Change (“A framework for nitrogen futures in the shared socioeconomic pathways”).

We agree that use of N-explicit scenarios would greatly reduce uncertainties on future river N projections. Since Kanter et al. (2020) suggest only narratives at this moment, we cannot use them to force the model yet. But, we have noted that this will be pursued in future work in the Discussion as:

Several limitations of our modeling framework should be considered when interpreting these results. First, while CMIP6 forcing datasets provide one of the most comprehensive future global land use, anthropogenic N, and CO₂ inputs to land biosphere models, they do not provide recycled N inputs. This limitation has been partly resolved by using the future manure applications constructed based on the CMIP6 forcing datasets^{31,32}. Our approach does not explicitly account for sewage, but its importance will be likely much smaller than manure applications²⁴. Ultimately, these uncertainties will be better explored when N-explicit scenarios, based on new N-focused narratives⁵⁵, become available to force land biosphere models.

4. Finally, it might be worth having a native English speaker review the paper to improve the flow of some sentences and increase the clarity of some messages.

Done. We feel that this additional review of the manuscript prose, along with the reviewer's constructive comments, has improved the clarity of the manuscript and the robustness of the results.

Specific comments:

p.1 lines 27-30: Highlight the other aspects of N pollution here, including multiple compounds and environmental human health impacts and its unique chemistry (i.e. the N cascade).

As suggested, we have extended the paragraph to highlight the other aspects of N pollution as:

Nitrogen (N) fertilizer usage and cultivation-induced biological N fixation (BNF) help feed nearly half the global population¹. These practices, in addition to fossil fuel burning for energy production, however, have increased reactive N losses to the environment, causing a cascade of negative impacts on ecosystem and human health^{2,3}. NO_x, N₂O, and NH₃ emissions to the atmosphere have contributed to acid rain⁴, air pollution⁵, stratospheric ozone depletion⁶, and the radiative forcing underlying climate change⁷. N fluxes from lands have also impaired freshwater quality⁸ and contributed to coastal eutrophication⁹, hypoxia¹⁰, and harmful algal blooms¹¹, putting aquatic resources and the communities that depend upon them at risk¹².

p.2 lines 42-44: This is a very general statement here. Make it clear that these terrestrial and freshwater ecosystem processes including other N losses to air and plant uptake.

We have attempted to address this comment and others by presenting all input/output and storage change fluxes with a new figure (Fig. 2), a new table (Table S5), and extended descriptions of N inputs and outputs in the Results. Please refer to our response to the reviewer's comment 2 for full details. The most relevant statements for this comment are on page 3, line 11 and page 3, line 27:

On page 3, line 11

While river DN load trends are aligned with fertilizer application trends, the projected magnitude and range of loads across the scenarios (34-43 TgN yr⁻¹ by the end of the 21st

century (2079-2099)) is far smaller than those of fertilizer applications (107-208 TgN yr⁻¹, Fig. 1, Table S5). Other inputs and processes in terrestrial and freshwater ecosystems clearly modulate the imprint of fertilizer applications on river DN loads. Relatively low inputs of BNF and atmospheric deposition (Fig. 2a, b) and high denitrification rates (Fig. 2c, d) damp river DN load increases in cases with relatively large fertilizer inputs (SSP1-2.6, SSP2-4.5), while the opposite holds for scenarios with low fertilizer additions (SSP5-8.5), reducing the gaps one would expect based on fertilizer alone.

On page 3, line 27

The positive outcomes in terms of water N pollution projected in SSP5-8.5, however, are likely accompanied by less favorable air pollution outcomes. Relatively high harvest (i.e., N in harvested woods, crops, and grasses) or net harvest (i.e., harvest after subtracting out recycled inputs, i.e., manure applied to croplands and grasslands, Fig. 2c, d), as well as high atmospheric deposition (Fig. 2a, b), in SSP5-8.5 is indicative of a high atmospheric N pollution potential. This is because much of the net harvest is likely to ultimately go to the atmosphere via wood, biofuel, and waste burning or emissions from food, human, and livestock waste. While N₂ emissions via this pathway are generally considered benign, and some harvested N is sequestered in durable goods (i.e., home building), a significant fraction would likely be emitted as NO_x, N₂O, and NH₃^{31,44}.

p.2 line 46: The low fertilizer additions aspect of SSP5-8.5 will be confusing to some readers who see it as a “worst case” scenario so make sure to explain this properly. This may also be the place to include mention of higher livestock production and thus manure.

We agree that the SSP5-8.5 result will run counter to the expectations of many readers. This is one of the reasons why we feel this result is particularly noteworthy. As described above, we now include an explicit consideration of manure, and we have expanded our discussion of the dynamics underlying our SSP5-8.5 result on page 3, line 18:

In the fossil fuel-intensive scenario SSP5-8.5, the demand and subsequent production of food and feed crops increase the most among the three scenarios considered herein (Fig. 3, Table S6). This occurs despite low population growth, reflecting the assumed prevalence of diets with high animal and waste shares^{13,39} (Table 1). The meat-rich diet also underlies the sharp increase in manure applications which peak in the 2040s, before decreasing after the 2060s due to limited grassland availability, especially in African and Latin American regions^{31,32}. Despite the highest food and feed crop production, projected fertilizer applications in SS5-8.5 increase only slightly (Fig. 1c) due to very low biofuel crop production in the absence of explicit climate mitigation policies (Fig. 3) and rapid

improvements in agricultural productivity^{13,39} (Table 1). River DN loads are projected to increase by only 7% (without manure applications, Fig. 1a) or 8% (with manure applications, Fig. 1b) from the present (2000-2020) to the end of the 21st century (2079-2099). This occurs despite the largest increase in atmospheric N deposition and BNF in SSP5-8.5 (Fig. 2a, b).

Reviewer #2 (Remarks to the Author):

Uneven pressure of global climate mitigation on regional water quality in the 21st century by Lee et al.

The authors are exploring three scenarios (SSP1, SSP2 and SSP5) and the impact on the dissolved nitrogen export to the ocean. They see that SSP5 leads to the lowest nitrogen export, and SSP2 the highest. Also they suggest that the high income countries (low GDP per capita) can meet their water N level and climate mitigation targets, but the low income countries do not meet both thresholds.

In this response, the line numbers correspond to those in the original manuscript before making any modifications. Unless noted, table and figure numbers in our responses correspond to the revised manuscript.

General remarks

I am thinking about the title. The authors conclude here that (line 6, 1-3) there is an insignificant relationship between precipitation and temperature with dissolved nitrogen export. So what do the authors mean by “global climate mitigation”?

The “climate mitigation” we are referring to here is the use of bioenergy to mitigate CO₂ emissions. This element of the mitigation scenarios SSP1-2.6 and SSP2-4.5 is critical to lowering the radiative forcing relative to the high emission scenario SSP5-8.5, but it requires fertilizer inputs. Thus, global climate mitigation efforts impact water quality. We have edited the title and the abstract to clarify the linkage between the title and the Results.

It is very hard to follow all the fluxes. I think I would like to have a kind of scheme with the following items for the three scenarios and for current situation and end of the century situation: total N input (and the different sources fertilizer, manure, N fixation and deposition), total N production (harvest and so on), N storage in soils, N load to the river, N export to the coast.

The requested input/output and storage change fluxes have been described in Fig. 2 and also added in a new Table S5, with explicit description text of the fluxes in the Results.

The model LM3-TAN is a complex model. The land processes are described a lot, but the river processes are still not clear to me. They need a good description. The validation of N in rivers is a major concern (see comments of Line 2, 8).

The detailed river processes can be found in the model description paper of LM3-TAN (Lee et al., 2014). We have extended the validation of N in rivers which is detailed in our response to the reviewer's comment *Line 2, 8*. We have also extended a description of the river processes in Methods as:

Terrestrial runoff of three N species (nitrate, ammonium, and DON) to river systems are subject to retention within rivers and lakes or transformed during transport to the coastal ocean^{20,29}. Each model grid cell contains one river reach and/or one lake, in which freshwater microbial processes (i.e., mineralization, nitrification, and denitrification) are simulated. First-order kinetics are used to describe the freshwater microbial processes. Reaction rate coefficients of freshwater mineralization and nitrification are prescribed. Reaction rate coefficients of freshwater denitrification are estimated by a reported nonlinear regression function based on Lotic Intersite Nitrogen experiment reach-scale measurements⁶³⁻⁶⁵. An Arrhenius-based relationship is used to adjust the rate coefficients for temperature effects, approximately doubling the rates for a temperature increase of 10°C. Water containing N in each river reach or lake flows to another river reach in the downstream grid cell following a network that ultimately discharges to the ocean. Freshwater physics, hydrology, and hydrography are described in detail elsewhere⁶¹.

One of the outcomes of this study is that SSP5 is the best scenario regarding water quality. This is an outcome that surprised me. Therefore I dived into the material (Hurt et al, 2020 and Popp et al. 2017) to see how the scenario were constructed. Fertilizer is the primary determinant of future nitrogen load. These are coming from Hurt et al. 2020. The used fertilizer is originated from three different models (SSP1: IMAGE, SSP2: MESSAGE and SSP5: MAgPIE. Background on these models can be found in Popp et al. 2017. There the following line is found: "Interestingly, the ranking within models is consistent, but the absolute values are strikingly different and mixing and matching models can be misleading." My question is here: is this comparison of three scenarios misleading, because it is based on three different models? Is it not better to take the three scenarios from one model?

The sentence the reviewer points out should be interpreted in the context of the entire paragraph and of the paper. Here we also copy several additional sentences prior to the sentence the reviewer copied from the paragraph.

“While much of the discussion in this article focused on comparing among marker scenarios, the dependence on these marker scenarios does not affect the qualitative conclusions in most cases. That is, using a single model across SSPs would not change the ranking of scenarios in most cases. One notable exception is in the interpretation of food prices under a mitigation case. Here, model differences dominate scenario differences. Interestingly, the ranking within models is consistent, but the absolute values are strikingly different and mixing and matching models can be misleading.”

It states that using a single model would not change the ranking of the scenarios and does not affect the qualitative conclusions or ranking in most cases, with an exception in the absolute values of food prices under a mitigation scenario. Our study does not interpret the absolute values of food prices.

We have shown with Fig. S6 (which is now Fig. 3) 1) generally higher fertilizer applications in the scenarios with higher biofuel crop production and 2) crop production ranges not only across the three scenarios, but also across all marker and non-marker models (AIM/CGE, GCAM4, IMAGE, MESSAGE-GLOBIOM, and REMIND-MAGPIE), demonstrating generally lowest crop production in SSP5-8.5. We clarify this further by reporting all crop production estimates within a new Table S6 which have been used to produce Fig. 3. The crop production estimates are available in public at SSP Database, <https://tntcat.iiasa.ac.at/SspDb/dsd?Action=htmlpage&page=40>.

Finally, we have expanded discussion of the factors contributing to lower fertilizer inputs in SSP5-8.5 to clarify the mechanisms and assumptions that underlie the SSP5-8.5 results, including apparent water and air pollution tradeoffs (please also refer to our response to the general comment 2 of reviewer #1).

The second point that surprised me is the low fertilizer use in SSP5. The reason for the low fertilizer use (of what I understand of this) is the fact that there is a meat-rich diet, which means that there is a lot of manure available to be used on agricultural land. I don't understand how this model is accounting for manure (at least I could not extract this from this article), but taking manure into account on a proper way could perhaps lead to a different conclusion. So this is also a major concern.

We acknowledge the primary concern of both reviewers regarding uncertainties associated with manure applications. We have identified nascent future manure application projections constructed by using the CMIP6 forcing datasets (Beaudor et al., 2023a, b). Since we have also

used the same CMIP6 forcing datasets for our projections, the manure applications can be and were integrated with our previous analysis.

The inclusion of the manure applications has increased global river DN loads only modestly and reinforced our primary findings, including that river DN loads are lowest globally and in high-income regions under SSP5-8.5. For SSP5-8.5, global river DN loads are projected to increase by 7% (without manure applications, Fig. 1a) or 8% (with manure applications, Fig. 1b) from the present (2000-2020) to the end of the 21st century (2079-2099). For SSP2-4.5, global river DN loads are projected to increase by 21% (without manure applications, Fig. 1a) or 25% (with manure applications, Fig. 1b).

The results of SSP1, SSP2-4.5, and SSP5-8.5 without manure applications and SSP2-4.5 and SSP5-8.5 with manure applications have been described throughout the manuscript. Please refer to our responses to the reviewer #1's comment *1* and *2* for details of a description of the manure applications and of most relevant results.

Remarks

Line x,y means page x, line y

Line 1, 28: "fertilizer runoff"?

We have modified it to "N fluxes from lands".

Line 1, 28 – 30: Very blunt to assign all impacts to fertilizer runoff. Change.

We have extended the paragraph as:

Nitrogen (N) fertilizer usage and cultivation-induced biological N fixation (BNF) help feed nearly half the global population¹. These practices, in addition to fossil fuel burning for energy production, however, have increased reactive N losses to the environment, causing a cascade of negative impacts on ecosystem and human health^{2,3}. NO_x, N₂O, and NH₃ emissions to the atmosphere have contributed to acid rain⁴, air pollution⁵, stratospheric ozone depletion⁶, and the radiative forcing underlying climate change⁷. N fluxes from lands have also impaired freshwater quality⁸ and contributed to coastal eutrophication⁹, hypoxia¹⁰, and harmful algal blooms¹¹, putting aquatic resources and the communities that depend upon them at risk¹².

Lines 1, 43 – 44: Here it is referred to 18 and 20. Both publications have considered the integrated effects. For 18 it is clear, because it is part of an integrate assessment model “IMAGE”. Number 20 is a set of regression equations. Basic data to feed this model are also from the IMAGE model.

We have modified the statement to clarify what we have intended to emphasize with this study as:

Projections to date, however, have focused on a subset of these drivers and have not investigated the integrated effects of interactions among atmospheric CO₂, climate change, and socioeconomic drivers shaping river N loads within a unified process-based framework^{17,24-26}.

Line 1, 45-46: Why is a projection to the mid -21 century giving a limiting perspective?

We are referring here to the fact that the projections only extend to the mid-21st century rather than continuing to the end of the 21st century, where the largest discrepancies between scenarios are generally found.

Line 2, 2: “most prior projections” and ending with two references which are an example. But why “most”? Give some examples of both cases.

Here we have listed the four most relevant river N projection studies based on 4 different classes of models: 11, 18-20. Please let us know if there are more projections by different classes of models.

Ref 20 (Global NEWS) assumes that land is in steady state. Ref 18 (IMAGE-GNM) assumes that natural lands are in steady state. Ref 11 uses an empirical relationship for projections. Ref 19 (SWAT) uses a process-based hydrology/water quality model for projections. Both models of Ref 11 and Ref 19 do not explicitly account for N storage changes in vegetation/soils for their projections, which puts these models outside of the scope of this discussion.

Based on the above description, we clarify the statement as:

Furthermore, most prior projections were derived from models that rely heavily on empirical relationships^{17,26} or assume that the N storage in river basins²⁶ or parts of river basins²⁴ is in steady state.

Line 2, 8: LM3-TAN is referenced by 14 and 23. 23 is only for part of the Mississippi but this is a global application. End of this paragraph, the validation is mentioned of this model. Figure S1 gives a very clear perspective. The discharge of DIN and DON are simulated too high. With a higher simulated discharge you would expect lower concentrations to correct this. But no, the model overestimates the concentrations as well. This means that this model has a problem and should be improved.

We have broken up this comment to ensure that each part is addressed.

We agree that simulated river DIN and DON loads were high. This is likely due in part to the change in forcing. Global climate models used to make projections can have significant biases in regional precipitation and temperature relative to observation-based atmospheric reanalysis products like those used by Lee et al. (2019). As the reviewer points out, however, some bias was evident even before the change in forcing. We thus recalibrated the model using 3 parameters which both are highly uncertain and impact mean river N load biases: the reaction rate constant for freshwater nitrification, the minimum reaction rate constant of freshwater denitrification, and the calibration factor for soil dissolved organic matter (details are provided below under Note S1). The values chosen fall within plausible ranges and largely removed the model biases (see Fig. S1). These steps are now described in Note S1.

We note that ref. 23 was for the Susquehanna River Basin, the largest of the basins in the northeastern U. S., and this is not a part of the Mississippi River Basin. While the paper is about the regional application, it serves as the base documentation paper for LM3-TAN, and was thus cited here. Ref. 14 provides the initial global application and historical validation which supports the current application.

Also the log transformation in the concentrations is not helping and this comparison should be done without log transformation.

The range of concentrations and loads vary over several orders of magnitude (e.g., range from 0.01 to 3.9 for nitrate concentrations) across globally distributed rivers. The log transformation provides a standard means of assessing the model's capacity to match concentrations and loads over the full range (e.g., Mayorga et al., 2010). That is, following the log-transform, a factor of 2 misfit for a base concentration of 0.01 mg/l is given the same weight as a factor of 2 misfit

around a value of 3.9 mg/l. Without the log-transform, higher values are given much greater weight than lower ones. Since our goal is a model that can capture globally-distributed systems with vastly different properties, we feel that the log-transform is most appropriate. We have, however, provided a comparison without log transformation for the reviewer's record (please see a figure at the end of this response).

More elaboration is needed why there is in DIN discharge a location with almost two orders difference. I also looked at the publication number 14. Also here the same problems occur. The discharge looks better in pub 14, but the concentrations are also here to high. What is the difference between the model used in pub 14 and the current model? Where is that described?

The primary difference is the forcing used for the two studies. Reference 14 used atmospheric forcing from an atmospheric reanalysis product that integrates observations and model results to reconstruct historical forcing over a relatively short period. The current application, however, is focused on climate change projections. This requires a global climate model capable of simulating the multidecadal evolution of the mean climate as it responds to changing greenhouse gasses, aerosols, and land use changes without any data assimilation (as there is no data to assimilate decades in the future). Global climate model projections are thus generally most skillful at continental scales and above (consistent with their intended global change applications), but can have significant regional biases (e.g., Randall et al., 2007). This is expected to result in some degradation in fit, but has the benefit of enabling projections into the future. Such degradation resulting from simulated climate forcings is also commonly shown in other river N modeling studies (e.g., R^2 decreases from 0.60 to 0.54 when switching observed hydrology to modeled hydrology in Fig. 3 of Mayorga et al. (2010)). We now clearly indicate this change in forcing relative to ref. 14 in Introduction (page 2, lines 8-16) and Note S1, in addition to Methods (page 7, lines 44-46).

Table S1 is, in another form (Table S2), also presented in that publication 14. A combination of 14 and table S1 should result in a new table S1 with load, discharge, concentration for DIN and DON with observations and simulations. Unclear for both, is how the simulated values are obtained. How many observations are the base of these observations?

As the reviewer requested, we have extended the Tables S1-S3 to include both measurement-based and LM3-TAN estimates for river water discharge, N loads and concentrations, with extended measurement-based data descriptions (please see the table captions for details in the data descriptions). We have also updated the measurement-based datasets from Meybeck and Ragu (1997) to the latest one, Meybeck and Ragu (2012), which is available in public at <https://doi.pangaea.de/10.1594/PANGAEA.804574>.

In terms of model evaluation, we have extended our cross-watershed evaluations by including analysis of additional N species (nitrate) for which a larger number of measurement-based river N estimates are available. This increased the total number of rivers analyzed to 53 rivers (Tables S1-S3). In addition to the Pearson correlation coefficient (r) between the log-transformed modeled and measurement-based estimates across the 53 rivers, we now also report the prediction error computed as the difference between the modeled and measurement-based estimates of loads expressed as a percentage of the measurement-based load (Table S4).

Regarding the derivation of simulated values, all the processes in LM3-TAN are updated every 30 minutes. This means that the model outputs (including river water and N throughout the river network) are given at the 30 minute time step. The annual results for water discharge, N loads and concentrations are the averages of the 30 minute results with applicable unit conversions. These high temporal resolution, continuous outputs do not require discharge-weighted calculations or any other additional result manipulations. Therefore, as shown in the following unit conversion equation,

$$\text{N concentration (mass/volume)} = \text{N load (mass)} / \text{Discharge (volume)}.$$

We have clarified this in the Methods section (page 8, line 7).

Why just an average (and I don't know which average that is) of the simulated values? I think this is a very minimal validation of this model. This dynamic model wants to capture future projections, but is poorly validated on just one time point. A dynamic model should be validated with long-term observations.

This study focuses on large-scale multi-decadal mean changes in river N in response to changes in climate, land use, fertilizers, and other inputs. Since data resolving changes at this scale are rare, a cross-system assessment of the model's capacity to capture contrasts between globally distributed watersheds sampling different climate, land use, and fertilizer/input characteristics provides an important, and relatively data-rich, validation step shared by all global river N load projection modeling studies. Indeed, we note that the level of river water discharge and N validations for future projections presented in this study is equal to or more than those of other future river N projections, for example, by the widely cited and pioneering model Global NEWS (Seitzinger et al., 2010), in that not only river N loads, but also river water discharge and N concentrations are evaluated against observations.

Over the past decade, we have augmented this core spatial comparison with a range of LM3-TAN comparisons against time varying data at various scales. Most notably, the riverine and land fluxes from LM3-TAN were found to be consistent with observation-based estimates, including multi-year to centurial changes in land C storage (ref. 14, added for the runs herein in

Note S1). In terms of evaluations against long-term observations from within a river basin, we have evaluated LM3-TAN against various rivers within the Susquehanna (Lee et al., 2014; Lee et al., 2016), Mississippi (Lee et al., 2021), and Korean Peninsula (Lee et al., 2018) River Basins, which are influenced by different climate, land use, and fertilizer applications. We note that such comparisons are difficult within climate change projections because simulated climate variability does not match the timing of observed climate variability. This is a standard in the design of climate projections, which are focused on multi-decadal climate change, and we thus lean on past studies to build confidence in the model's capacity to reasonably represent climate, land use, and fertilizer-driven variability within a watershed.

We now have noted the forcing change, extended evaluation, and associated results in the Introduction (page 2, lines 8-16) as:

In this study, we use the process-based, terrestrial-freshwater ecosystem model LM3-TAN^{20,29} to project spatially explicit, global river N loads over the 21st century. While LM3-TAN has been implemented at relatively coarse 1-degree resolution and does not resolve river water management (e.g., dams), it simulated global and regional terrestrial and freshwater carbon (C) and N storage and fluxes to the ocean and atmosphere consistently with published synthesis from 28 different studies^{18,20}. With necessary updated forcings and minor calibration refinements to make future projections herein (see Methods, Note S1), LM3-TAN is found to accurately capture observed water discharge, nitrate, dissolved inorganic N (DIN), and dissolved organic N (DON) concentrations and loads from 53 large rivers influenced by various climate and socioeconomic conditions, with skill comparable to empirical approaches while maintaining consistency with global soil N storage, terrestrial C storage changes, and river DIN and DON load estimates (Note S1, Tables S1-S4, Fig. S1). LM3-TAN has also simulated realistic long time series N load variability in several river basins^{18,29}.

The model evaluations have been extensively described within a newly added Note S1 as:

Note S1. Model performance analysis.

For cross-watershed evaluations, LM3-TAN results of river water discharge (m^3/s), nitrate, dissolved inorganic N (DIN), and dissolved organic N (DON) loads (ktN/yr) and concentrations (mgN/l) are compared with measurement-based estimates from 53 of the world's major rivers (Tables S1-S3)^{1,2}. We report the Pearson correlation coefficient (r) between the log-transformed modeled and measurement-based estimates across the 53 rivers and the prediction error computed as the difference between the modeled and measurement-based estimates of loads expressed as a percentage of the measurement-based load (Table S4). The river results are presented for 1) the year 1990, 2) the range by using annual results for the period 1990-2000, and 3) the 1990-2000 average.

With necessary changes in the model forcings to make future projections (see Methods), we note three parameter changes from the global implementation of LM3-TAN³. The three parameters are 1) the reaction rate constant for freshwater nitrification^{3,4} from 0.51 to 0.1 within a reported range of 0.04-3 1/day in Bowie et al. (1985) and references there⁵ 2) the minimum reaction rate constant of freshwater denitrification^{3,4} from 0.034 to 0.18 within a reported range of 0.034-117 1/day in Alexander et al. (2009)⁶, and 3) the calibration factor for soil dissolved organic matter^{3,4} introduced in Lee et al. (2014)⁴ from 1 to 150.

With this minor calibration refinements, LM3-TAN is found to accurately capture observed water discharge, nitrate, DIN, and DON concentrations and loads from 53 large rivers influenced by various climate and socioeconomic conditions with skill comparable to empirical approaches (Tables S1-S4, Fig. S1). We note the degradation of fit relative to the previous global implementation of LM3-TAN³. The primary difference is the forcings used for the two studies. The previous study used atmospheric forcing from an atmospheric reanalysis product that integrates observations and model results to reconstruct historical forcing over a relatively short time period. The current application, however, is focused on climate change projections. This requires a global climate model capable of simulating the multidecadal evolution of the mean climate as it responds to changing greenhouse gasses, aerosols, and land use changes without any data assimilation (as there is no data to assimilate decades in the future). Global climate model projections are thus generally most skillful at continental scales and above (consistent with their intended global change applications) but can have significant regional biases⁷. This is expected to result in some degradation in fit, but has the benefit of enabling projections into the future. Such degradation resulting from simulated climate forcings is also commonly shown in other river N modeling studies (e.g., R² decreases from 0.60 to 0.54 when switching observed hydrology to modeled hydrology in Fig. 3 of Mayorga et al. (2010)⁸).

For global evaluations, LM3-TAN results of river DIN and DON loads and soil N storage are compared with published estimates⁸⁻¹¹. Global amounts of river DIN and DON loads are found to be consistent with published ranges (Table S4). In addition, simulated global soil N storage during the 1980s-1990s are 86 PgN and 86-87 PgN without and with manure applications, which are within published ranges of 95 (70-820) PgN/yr in Post et al. (1985) and references there¹⁰.

Although observations of global and large regional land N storage and flux changes over long time periods are not available for model validation, global terrestrial carbon (C) changes are often alternatively used to evaluate those of N given the closely coupled C and N cycles³. LM3-TAN results of global terrestrial C storage and flux changes on centennial to multi-year time scales are compared with estimates from large-scale constraints and atmospheric studies¹²⁻¹⁵. Simulated global net land C fluxes are 1.8 PgC/yr and 2.0 PgC/yr without and with manure applications for the 1990s and 1.8 PgC/yr and 1.8 PgC/yr for

2001–2004, in consistent with estimates based on global C budget constraints (e.g., 1.1 (0.5–1.8) PgC/yr for the 1990s)¹³ and inverse models (e.g., 0.3–1.7 PgC/yr for 2001–2004)^{14,15}, albeit at the upper end. The simulated cumulative net land C source between 1750 and 2011 is 21-23 PgC (this range is due to the climate forcing differences across the three scenarios from 2005, see Methods) without manure applications and 3 PgC with manure applications. Both cases without and with manure applications are consistent with a reported uncertainty range of 30 ± 45 PgC between 1750 and 2011¹². The simulated land-use change contribution to elevated CO₂ between 1750 and 2011 is 188-189 PgC (this range is due to the climate forcing differences across the three scenarios from 2005, see Methods) without manure applications and 176-177 PgC with manure applications. Both cases without and with manure applications are within a published range of 180 ± 80 PgC between 1750 and 2011¹².

Lastly, we note that several previous studies have demonstrated the capacity of LM3-TAN to reasonably represent temporal variability within the Susquehanna^{4,16}, Mississippi¹⁷, and Korean Peninsula¹⁸ River Basins, which are influenced by different climate, land use, and fertilizer applications. The validated results have shown simulated vegetation and litter/soil N storage changes to significantly alter seasonal to multi-year river N extremes^{16,17}.

Randall, D. A. et al. *In Climate Change 2007: The Physical Science Basis* (eds Solomon, S. D. et al.) Climate Models and Their Evaluation (Cambridge University Press, Cambridge, United Kingdom and New York, NY, USA, 2007).

Lee, M., Malyshev, S., Shevliakova, E., Milly, P. C. D. & Jaffé, P. R. Capturing interactions between nitrogen and hydrological cycles under historical climate and land use: Susquehanna Watershed analysis with the GFDL Land Model LM3-TAN. *Biogeosciences* **11**, 5809–5826 (2014).

Lee, M., Shevliakova, E., Malyshev, S., Milly, P. C. D., & Jaffé, P. R. Climate variability and extremes, interacting with nitrogen storage, amplify eutrophication risk. *Geophys. Res. Lett.* **43**, <https://doi.org/10.1002/2016GL069254> (2016).

Lee, M. et al. Control of nitrogen exports from river basins to the coastal ocean: Evaluation of basin management strategies for reducing coastal hypoxia. *J. Geophys. Res. Biogeosci.* **123**, 3111–3123 (2018).

Lee, M., Stock, C. A., Shevliakova, E., Malyshev, S. & Milly, P. C. D. Globally prevalent land nitrogen memory amplifies water pollution following drought years. *Environ. Res. Lett.* **16** 014049 (2021).

Line 3, 9 In Figs S3-S5 the temperature is 5.3. Here 6.0. Which one is correct?

The 5.3°C increase is over the entire globe and the 6.0°C increase is over land. This has been described in the Methods (page 9, lines 39-41).

Line 7, 5: Can you be clear what water N level you mean and what value is used?

We have clarified it as “decreasing or stabilizing water N loads relative to current levels”.

Line 9, 30: “20% decrease of NUE”. How is that done? From the first scenario year 20%? The same question for precipitation (line 9, 36) and air temperature (line 9, 39). Which year is the first scenario year in this study?

Since we now include simulations that apply manure applications, we have removed the NUE sensitivity simulations.

The precipitation and temperature changes within the climate sensitivity simulations have been described in Figs. S2-S4. These simulations were run from the year 2020 which we have clarified in the Methods (page 9, lines 36-41) as:

- 1) Precipitation in SSP5-8.5 was proportionally increased to yield a 7.6% (8.2% over land) increase by 2080-2099 relative to 1986-2005. This simulation run from the year 2020 evaluates an impact of the upper end of precipitation increases from multi model simulations (IPCC, 2007)⁴³.**
- 2) Air temperature in SSP5-8.5 was proportionally increased to yield a 5.3°C (6.0°C over land) increase by 2080-2099 relative to 1986-2005. This simulation run from the year 2020 evaluates an impact of the upper end of temperature increases from multi model simulations (IPCC, 2014)⁴⁴.**

Line 17, : Figure 3: The SSPs are developed as storylines. Population and GDP were the first parameters that were available. From this starting point all activities are calculated. Some of the parameters in the SSPs are determined based on GDP development. My question is whether fertilizer and atmospheric deposition is created based on direct or indirect GDP development or not? The answer determines whether this figure should be include or not.

The population and GDP projections are key scenario drivers used for the development of IAM scenarios that elaborate the SSPs in terms of energy, land use, and emissions. Fertilizer applications and atmospheric deposition are driven by the IAM SSP scenarios of energy system, land use changes, and resulting air pollutant, greenhouse gas emissions, atmospheric concentrations, crop production, yield-increasing technology, etc (Riahi et al., 2017; Popp et al., 2017; Hurtt et al., 2020; Rao et al., 2017; Durack, et al., 2021). Thus, fertilizer applications and atmospheric deposition cannot be independent of pcGDP.

Supporting information

Table 1: last line in caption. “Log 10 of discharges” Which unit is used here?

The unit is m³/s, which now we have clarified in the table caption.

Figure S2a: Changes in soil and freshwater storage. I don’t understand what is included in this. Can you make this more clear (without referring to Lee et al. (2021))

We have added a description of the storage in the figure (Fig. 2) caption as:

The terrestrial and freshwater storage includes 5 vegetation storage (leaves, fine roots, sapwood, heartwood, and labile storage), 4 organic soil storage (fast and slow litter, slow and passive soil), 2 inorganic soil storage (ammonium and nitrate), and 6 freshwater storage (river DON, ammonium, and nitrate, lake DON, ammonium, and nitrate).

Figure S2a: Is this really 20 year moving average? Looks to me that it is not.

Yes, they are 20 year moving averages. We now have presented 21 year moving averages, so that the center of an average period (e.g., 2000-2020) can be placed in the x-axis (e.g., 2010).

Figure S3 and S4: Explain the regions again.

We have added a region description in the figure captions.

Concentration results without log transformation for the reviewer's record. The first line is the results without manure applications and the second line is the results with manure applications.

REVIEWER COMMENTS

Reviewer #1 (Remarks to the Author):

The authors have satisfactorily addressed all my comments. I am happy to recommend this article for publication.

Reviewer #2 (Remarks to the Author):

Uneven pressure of global climate mitigation on regional water quality in the 21th century by Lee et al., revision 1 by reviewer 2.

First of all many thanks on all the improvements that are already made based on the comments of both reviewers. That is appreciated.

However, the current version of the manuscript is not yet suitable for publication.

General comments

Both reviewers mentioned the role of manure in the scenarios. I am happy that the authors tried to incorporate the manure into their calculations. However, the scenario database they used is not published. I tried to find the background of these manure calculations, but the website mentions: "Beaudor, M., N. Vuichard, J. Lathière, D. Hauglustaine., Historical and future ammonia emissions database (2000-2100) from the CAMEO process-based model, in preparation." I could not find the assumptions they made to produce these datasets. I think that this dataset can not be used, as long as it is not published!

Next to this, I think the authors should use a database which consist of the three scenarios. Especially, because the discussion is between SSP5 and SSP1 and SSP1 is not provided!

Thirdly, I think that the main text should only include scenarios with manure included. The manure should be part of the discussion. This makes the manuscript more readable. Now a new reader could think that manure is one of the options in the scenarios. But this is not the case. But this is a suggestion.

I am happy with an overview of the fluxes (Table S5). I am missing the units (I assumed that it is in Tg N/yr). But two extra questions are raised by this table. 1. I understand that the nitrogen part in the harvest is higher, when the nitrogen input is higher. But is this increase reasonable? So apparently 6 Tg is going to the harvest of the 63 Tg manure? This is 10%. Can you elaborate on this? I also see that in Figure 3, that the crop production in t DM is higher with manure. Is the production not determined by the feed/food demand? 2. The BNF is going down with around 20 Tg when the manure is incorporated. So one third of the manure is replacing the BNF? It is known that the BNF is reduced, when the plant has enough nitrogen to use. But here 8-9% of the total BNF is reduced. I tried to find how you calculate the BNF and I could not find it. Can you elaborate on this?

Table S6: what is the unit?

Thanks for the information on the validation of the model. It is nice to see the tables S1 – S3. I think that additional work should be done to improve the validation of the model for at least the major river basins. But I can understand that the authors don't want to do extra work for this...

Reviewer #2 (Remarks to the Author):

Uneven pressure of global climate mitigation on regional water quality in the 21st century by Lee et al., revision 1 by reviewer 2.

First of all many thanks on all the improvements that are already made based on the comments of both reviewers. That is appreciated. However, the current version of the manuscript is not yet suitable for publication.

We thank the reviewers for their constructive comments throughout the review process. As we detail below, we have worked to thoroughly address the reviewer's remaining concerns.

General comments

Both reviewers mentioned the role of manure in the scenarios. I am happy that the authors tried to incorporate the manure into their calculations. However, the scenario database they used is not published. I tried to find the background of these manure calculations, but the website mentions: "Beaudor, M., N. Vuichard, J. Lathière, D. Hauglustaine., Historical and future ammonia emissions database (2000-2100) from the CAMEO process-based model, in preparation." I could not find the assumptions they made to produce these datasets. I think that this dataset can not be used, as long as it is not published!

The future manure application dataset has been published at <https://zenodo.org/records/10100435>, with a citation of the associated paper (Beaudor et al., 2023, GMD) that describes the methodology used and assumptions made to estimate the manure production and applications. In this revision, we have also fully entrained the lead authors of the study providing the manure dataset and added a paragraph that describes the necessary updates to model inputs required for future projections of manure applications in the Methods as:

Page 10, lines 23-26

Manure applications constructed by using the CMIP6 forcing datasets were taken from the dynamical agricultural module (CAMEO, Calculation of AMmonia Emissions in ORCHIDEE) within the land surface model ORCHIDEE³⁹⁻⁴¹.

Page 10, lines 38-46, page 11, lines 1-4

Manure applications are available for the historical period 2002-2014 and for the future period 2015-2100 for the SSP2-4.5, SSP4-3.4, and SSP5-8.5 scenarios, with SSP4-3.4 and SSP5-8.5 respectively representing the lowest and highest pathways of livestock production and manure applications among the CMIP6 IAM scenarios^{40,41,72}. The process-based model CAMEO, which was developed and validated in Beaudor and colleagues³⁹, was used to estimate manure dynamics. In order to project future manure production and applications,

CAMEO was run with updated inputs. The updated inputs are climate forcings taken from the Earth System Model IPSL-CM6A-LR⁷³, the same CMIP6 forcing datasets as used in this study (i.e., land use³⁰, fertilizer applications³⁰, atmospheric deposition³¹, and livestock production⁷²), and reconstructed gridded future livestock densities. Future livestock production provided for large regions⁷² was downscaled to the grid-cell level by assuming that grass-feed livestock demand was met locally, based on future trends in grassland areas, above-ground net primary productivity of the grassland, and a Grazing Intensity parameter introduced in Beaudor and colleagues³⁹.

In light of the substantial new data and text contributions of the scientists who led the future manure application projections, they have met the requirements for co-authorship and agreed to be added as co-authors. Finally, additional details of the future manure applications are available as a pre-print in Journal of Advances in Modeling Earth Systems at <https://doi.org/10.22541/essoar.170542263.35872590/v1>, which we have also included in accordance with reference guidance of Nature Communications. We emphasize, however, that the entire methodology of estimating manure production and applications is described in the fully published paper (Beaudor et al., 2023, GMD) and the leads of this work are now co-authors of our manuscript.

Finally, we would like to note that, to our knowledge, the future manure applications that we have used in this study are the only currently available documented projections for the entire 21st century. To project river N loads for the period 2015-2050, Beusen et al. (2022) also used future manure projections (with a citation of van Vuuren et al. (2017)), but this manure dataset documentation was not published in van Vuuren et al. (2017) or, to our knowledge, elsewhere. We thus cannot use this alternative. We feel, however, that the combination of the published methodology (Beaudor et al., 2023, GMD), the published dataset (<https://zenodo.org/records/10100435>), the details now included herein, the co-authorship of the dataset developers, and the pre-print with further details provide substantial documentation.

Beusen, A. H., Doelman, W. J. C., Van Beek, L. P. H., Van Puijenbroek, P. J. T. M., Mogollón, J. M., Van Grinsven, H. J. M., Stehfest, E., Van Vuuren, D. P. & Bouwman, A. F. Exploring river nitrogen and phosphorus loading and export to global coastal waters in the Shared Socio-economic pathways. *Glob. Environ. Change* **72**, 102426 (2022).

Next to this, I think the authors should use a database which consist of the three scenarios. Especially, because the discussion is between SSP5 and SSP1 and SSP1 is not provided!

Unfortunately, there is no such database. If documented future manure application projections were available for SSP1-2.6, we would use them. In lieu of simply discussing the paucity of data, we have entrained the scientists working at the frontier of this research area to provide the most rigorous possible evaluation of the sensitivity of our results to manure applications. While imperfect, the resulting best possible analysis substantively supports the primacy of fertilizer applications in determining future water quality pathways in a number of ways:

First, our paper now clearly shows that the magnitude of manure applications and the magnitude of the differences in manure applications between the lowest and highest manure application scenarios is far less than the magnitude of fertilizer applications and the magnitude of the differences in fertilizer applications between the scenarios (modified Fig. 1c). We feel that the stark visual on its own offers compelling evidence for the importance of fertilizer pathways over manure in determining water quality futures. We now augment this further with a comparison of the projected livestock production (newly added Fig. S5), a primary determinant of manure production, showing that SSP1-2.6 slots in between the “low manure” limit and SSP2-4.5 and would thus generate similar second-order variations in manure inputs. For the editor’s and reviewer’s convenience, we have added these figures at the end of this response.

Second, running SSP2-4.5 and SSP5-8.5 with and without manure shows that, as expected from Fig. 1c, the small differences in manure inputs relative to the differences in fertilizer inputs have generated commensurately small shifts in the relative water quality implications of each scenario that do not shift the order imposed by the stark differences in fertilizer applications (Fig. 1a, b, Figs. 4-5). The only inference required is that similar secondary differences in manure applications between SSP1-2.6 and the other scenarios (firmly supported by Fig. 1c and Fig. S5) would lead to similar secondary shifts in river N loads. It certainly would not eliminate the importance of considering the very large differences in fertilizer outputs between scenarios.

The reviewers’ request that initiated our “manure efforts” was to assess the sensitivity of our results to its inclusion. While we agree that the perfect solution would have been the availability of detailed manure projections for all scenarios, we have gathered the best available data to provide substantive insight into this issue, with one notable caveat (page 7, lines 38-46).

Furthermore, we emphasize that the now substantive analysis of the sensitivity of results to manure is just one of several novel sensitivities exploring N pollution pathways in this study. We have also demonstrated significant terrestrial and freshwater N responses to concurrently changing drivers (e.g., CO₂ fertilization feedbacks, biological N fixation feedbacks, etc.) that were omitted by prior studies.

Lastly, while uncertainties in future N forcing and earth system responses remain, the large differences in fertilizer inputs and the roots of these differences (i.e., food/feed and biofuel crop demands, climate mitigation, and agricultural technologies, practices, and management) are important to acknowledge and understand the important implications even with some uncertainty in manure applications for SSP1-2.6. That is, prominent climate mitigation/water quality tradeoffs exist, both at the global and regional scales, even if some gaps in our understanding of manure applications remain. We have endeavored to emphasize this very robust finding throughout the paper. In particular, we now close our conclusion with the statements:

While uncertainties in N inputs across future scenarios and the earth system response to these inputs remain, our results highlight the need for careful consideration of unintended, adverse effects of bioenergy-driven climate mitigation efforts on water quality and the inequity of outcomes along a socioeconomic spectrum. Projections in high-income regions provide cause for optimism for meeting climate mitigation targets and decreasing or stabilizing water N loads relative to current levels, but a commitment to rapid transfer of agricultural advances to low-income regions and bioenergy production options without raising fertilizer applications⁶⁰ may be required to meet these goals across the economic spectrum.

Thirdly, I think that the main text should only include scenarios with manure included. The manure should be part of the discussion. This makes the manuscript more readable. Now a new reader could think that manure is one of the options in the scenarios. But this is not the case. But this is a suggestion.

We agree that it would be ideal to have full manure datasets for all scenarios. We would use these if they were available. However, in their absence, we have been limited to providing the most rigorous sensitivity analysis possible. As described above, this sensitivity supports our main results and provides an avenue for highlighting the need for continued advances in projections of manure and other water quality relevant quantities. We have also worked to better integrate the Results and Discussion associated with the manure into the revised manuscript. In short, manure sensitivity is systematically presented along with CO₂ fertilization as a primary sensitivity, and discussed accordingly.

In the Introduction (**page 2, lines 38-42**), we have added a clear description of the manure simulations as:

We also explore the robustness of our findings to recently developed manure application projections³⁹⁻⁴¹. While these projections are not available for SSP1-2.6, scenarios for SSP2-4.5 and SSP5-8.5 and a scenario with the lowest livestock production and manure

applications (SSP4-3.4) provide a means of comparing the relative importance of cross-scenario fertilizer and manure application differences.

In the Results and Discussion, we have noted the prominent role of fertilizers on water quality futures, the robustness of our primary findings to manure applications, and the needs for additional assessment of manure dynamics.

On page 3, lines 15-23:

Differences in manure applications between the scenarios are considerably smaller than the differences in fertilizer applications, even when the “lower bound” scenario is considered (Fig. 1c). Manure applications in SSP5-8.5 are higher than those in SSP2-4.5 until the 2060s, but declines by the end of the 21st century and does not significantly shift the high fertilizer-driven river DN loads in SSP2-4.5 relative to SSP5-8.5 (Fig. 1b). While manure applications for SSP1-2.6 are unavailable, relative patterns in livestock production (Fig. S5), a primary determinant of manure production³⁹⁻⁴¹, suggest that manure applications in SSP1-2.6 lie between the lower bound scenario and SSP2-4.5. This would again yield differences in N inputs due to manure that are far less than cross-scenario fertilizer contrasts.

On page 4, lines 35-45:

In the sustainability scenario SSP1-2.6, low population growth, low-meat diets with low food waste, and advances in agricultural systems limit growth in fertilizer demand for food and feed crop production, but these factors are countered by biofuel-driven fertilizer demand associated with ambitious climate mitigation efforts^{13,35} (Table 1, Fig. 3). Strong air pollutant controls, high shares of renewables, and rapid energy efficiency improvements decrease atmospheric deposition markedly (Fig. 2a, b)^{35,46}, but not by enough to avoid moderately increasing N inputs (Fig. 2e) leading to a 14% river DN load increase that lies between the SSP5-8.5 and SSP2-4.5 responses (Fig. 1a). **Given the relatively low livestock production in SSP1-2.6 (Fig. S5), manure applications would likely enhance this increase by a similarly modest or lesser value than the 1-4% enhancements projected by SSP5-8.5 and SSP2-4.5, yet would not be enough to shift the order imposed by the stark differences in fertilizer applications.**

On page 5, lines 40-44:

As in the global case, additions of manure applications to SSP2-4.5 and SSP5-8.5 did not impact the trend contrasts between low and high pcGDP regions (Figs. 4-5). While manure effects on SSP1-2.6 are uncertain, the secondary importance of these inputs relative to fertilizers (Fig. 1c) limits their impacts and even partly compensating responses would not eliminate equity concerns imposed by fertilizer trends.

On page 7, lines 38-46:

Several limitations of our modeling framework should be considered when interpreting these results. First, while CMIP6 forcing datasets provide one of the most comprehensive future global land use, anthropogenic N, and CO₂ inputs to land biosphere models, they do not provide recycled N inputs. This limitation has been partly resolved by using the future manure applications constructed based on the CMIP6 forcing datasets³⁹⁻⁴¹. **Although the current scientific literature does not provide documented future manure applications for SSP1-2.6, the results based on the available manure projections suggests they are secondary to fertilizers both in the overall magnitude of the inputs and the contrasts between scenarios. It would be, however, best to have an explicit manure estimate for SSP1-2.6, and additional assessment of manure dynamics is needed.**

I am happy with an overview of the fluxes (Table S5).

Thank you, we have broken up this comment to ensure that each part is addressed.

I am missing the units (I assumed that it is in Tg N/yr).

Yes, the unit is TgN/yr for fluxes and TgN for storage, which we now have clarified in the table caption.

But two extra questions are raised by this table. 1. I understand that the nitrogen part in the harvest is higher, when the nitrogen input is higher. But is this increase reasonable? So apparently 6 Tg is going to the harvest of the 63 Tg manure? This is 10%. Can you elaborate on this?

To our knowledge, the contribution of manure applications to harvest has not been explored or measured at global scales, and therefore unknown. This makes it difficult to claim whether any given global estimates are reasonable or not.

As the reviewer has noted below, the manure applications lead to the BNF reductions in our simulations. Thus, the difference of total N inputs to soils between the simulations with and without manure applications is ~44 Tg/yr, rather than the total amount of manure applications, ~63 Tg/yr. In case of 2000-2020 averages for SSP2-4.5 with vs. without manure applications, for example, we estimate that the additional input to soils (44 Tg/yr) is lost to the atmosphere via soil denitrification and fire (16 Tg/yr, 36% of 44 Tg/yr), stored within terrestrial vegetation and soils (12 Tg/yr, 27%), leached to freshwaters (10 Tg/yr, 23%), and exported as harvest (6 Tg/yr 14%). In our simulations, ~70% of the manure is directly applied to pasture, and the other ~30% is applied to cropland as fertilizers. This, in part, explains the large losses to the environment (air

and water, 59%) and relatively inefficient uptake by grasses and its exports via livestock (i.e., grazing).

In this revision, we now have added soil denitrification estimates in Table S5.

I also see that in Figure 3, that the crop production in t DM is higher with manure. Is the production not determined by the feed/food demand?

As described in page 3, lines 40-44, fertilizer applications in the scenarios depend on food/feed and 2nd generation biofuel crop demands and agricultural productivity. In LM3-TAN, crop production is simulated based on various factors, including fertilizer applications, land use changes, and climates (Shevliakova et al., 2009). Our results demonstrate that 30% of the manure applied to cropland as fertilizers enhances the crop production as demonstrated in Fig. 3.

2. The BNF is going down with around 20 Tg when the manure is incorporated. So one third of the manure is replacing the BNF? It is known that the BNF is reduced, when the plant has enough nitrogen to use. But here 8-9% of the total BNF is reduced. I tried to find how you calculate the BNF and I could not find it. Can you elaborate on this?

To our knowledge, the contribution of manure applications to downregulation of BNF has not been explored or measured at global scales, and therefore unknown. The physiological process-based BNF modeling approach of LM3-TAN, however, suggests down-regulation of BNF when additional N via manure recycling is available for plant uptake. This result is consistent with current literature (e.g., Barron, 2007). The BNF modeling description was documented in detail within the terrestrial component description paper of LM3-TAN (Section 2.6 Biological Nitrogen Fixation in Gerber et al. (2010)).

We have added these points with the following statements in the Table S5 caption:

The physiological process-based BNF modeling approach of LM3-TAN suggests down-regulation of BNF when additional N via manure applications is available for plant uptake, consistent with the current literature (e.g., Barron, 2007). While the recycled inputs (i.e., manure applications) reduce BNF and thus reduce total new N inputs into the land systems (Fig. 2), they increase total N inputs to soils which lead to modest increases in harvest.

Barron, A. R. Patterns and controls of nitrogen fixation in a lowland tropical forest, Panama. Ph.D. diss., Princeton University, Princeton, NJ, USA (2007).

Table S6: what is the unit?

Thank you for pointing that out. The unit is t DM/yr, which we now have clarified in the table caption.

Thanks for the information on the validation of the model. It is nice to see the tables S1 – S3. I think that additional work should be done to improve the validation of the model for at least the major river basins. But I can understand that the authors don't want to do extra work for this...

In our previous response, we augmented the model validation by extending cross-watershed evaluation with an additional N species (nitrate) for which a larger number of measurement-based estimates are available and by including global evaluation of soil N storage, river N loads, and changes in land C storage and fluxes over multi-year to centurial scales. This additional validation builds on that provided in previous studies (e.g., Lee et al., 2019). While we are always open to developing additional diagnostics, we feel that the ones provided herein are those which are most germane to the primary conclusions of this paper (e.g., the large-scale river N load responses to large contrasts in inputs across scenarios on multi-decadal mean time scales, the magnitude and nature of potential compensating BNF and/or CO₂ fertilization effects).

We have added Fig. 1 for the reviewer's and editor's convenience. Panel c highlights the relative size of the differences in fertilizer inputs and manure inputs. While a manure projection for SSP1-2.6 is not available, we do have the lower bound scenario, which is indicated by the gray dashed line. This clearly shows that the differences in fertilizer applications, particularly beyond the early 21st century, are much larger than the differences in manure applications.

Fig. 1. Rising global river N loads and fertilizer applications. **a, b,** River dissolved inorganic N (DIN), dissolved organic N (DON), and dissolved N (DN, the sum of DIN and DON) loads to the coastal ocean for simulations without manure applications (**a**) and with manure applications (**b**). **c,** Fertilizer and manure N applications. All plots show 21-year moving averages from 2000 to 2099. The range across three scenarios (indicated by colors) provides a first-order estimate of uncertainty due to alternative socioeconomic developments and climate policies. The uncertainty of results to future CO₂ fertilization effects are demonstrated by comparing the marker scenarios (solid lines) with those that exclude CO₂ fertilization (dotted lines). See Methods for a detailed description of model simulations.

We have also added Fig. S5 for the reviewer’s and editor’s convenience. Livestock production is a primary determinant of manure applications. This figure compares livestock production in SSP1-2.6 to the other projections for which manure projections are available. While livestock production in SSP1-2.6 is lower than SSP2-4.5 and SSP5-8.5, it is elevated relative to the “low manure” limit.

Fig. S5. Livestock production. These four IAM marker scenario projections are provided from the SSP Database²⁵ for the period 2005-2100.

REVIEWER COMMENTS

Reviewer #2 (Remarks to the Author):

Uneven pressure of global climate mitigation on regional water quality in the 21th century by Lee et al., revision 2 by reviewer 2.

First of all many thanks for all the efforts the authors did to improvement this manuscript.

I am happy with the preprint version of the future manure applications. Although this is not a published paper, I found in this preprint the assumptions needed to compute the future manure applications.

I don't agree with the statement of the authors:

"The future manure application dataset has been published at <https://zenodo.org/records/10100435>, with a citation of the associated paper (Beaudor et al., 2023, GMD) that describes the methodology used and assumptions made to estimate the manure production and applications."

The GMD describes the methodology used, but did not describe the assumptions to estimate the future manure production and application.

I leave it to the editor, whether it is allowed to refer to a preprint.

I am satisfied with the adjustments and replies of the authors!

Reviewer #2 (Remarks on code availability):

The code contains only source code without any input data.

The README is not available, so no installation guide or how to run the application is not available.

So it is not possible to reproduce the results based on this source code.

Reviewer #2 (Remarks to the Author):

Uneven pressure of global climate mitigation on regional water quality in the 21th century by Lee et al., revision 2 by reviewer 2.

First of all many thanks for all the efforts the authors did to improvement this manuscript.

I am happy with the preprint version of the future manure applications. Although this is not a published paper, I found in this preprint the assumptions needed to compute the future manure applications.

I don't agree with the statement of the authors:

“The future manure application dataset has been published at <https://zenodo.org/records/10100435>, with a citation of the associated paper (Beaudor et al., 2023, GMD) that describes the methodology used and assumptions made to estimate the manure production and applications.”

The GMD describes the methodology used, but did not describe the assumptions to estimate the future manure production and application.

I leave it to the editor, whether it is allowed to refer to a preprint.

I am satisfied with the adjustments and replies of the authors!

We are grateful to the reviewers for their time, energy, and incisive comments throughout all stages of the review process. We believe that they have helped improve our manuscript substantially.

Reviewer #2 (Remarks on code availability):

The code contains only source code without any input data.

The README is not available, so no installation guide or how to run the application is not available.

So it is not possible to reproduce the results based on this source code.

Reviewer 2 points out that, while we have provided the full source code, the forcing/input files and platform-specific installation guide have not been provided. We have carefully considered this comment, together with the “Guidelines for authors submitting code & software”. We have questions about the applicability of reviewer 2’s request to our paper and the tractability of the guidelines for global land surface models (as described in Buechel, 2021).

Regarding the request for all of the model input data, the simulations require high-frequency (8 times daily) atmospheric forcing data totaling 899 GB and other inputs totaling 362 MB (Please see a data description in a newly added Table S7). This is not attachable with a zip file along with this manuscript and it greatly exceeds the ~50 GB limits of commercial data storage services like Zenodo.

We have worked to address this challenge in three ways. First, Table S7 provides a description of all of the used inputs and links to all of the datasets that are publically available. Second, additional inputs that are small enough (< 25 MB) have been directly uploaded from <https://github.com/minjinl/LM3-TAN>, along with the full model source code. Third, the high-frequency atmospheric forcing data and other large inputs, which are stored on NOAA/GFDL's institutional archive, can be provided upon request through the Globus high capacity data transfer service. This standard seems to have been applied in cases similar to ours (e.g., Bottino et al., 2024), though we are open to any alternatives the journal recommends. We have clarified all of these elements in the section on "Data availability" in accordance with Nature journal guidelines.

Regarding the requests for code installation/software guide, we would like to first point out that this study is an application of previously published code/model, including prior publication in Nature Communications (Lee et al., 2014; Lee et al., 2019). The "Guidelines for authors submitting code & software" of Nature journals state:

"If the custom algorithm or software is central to the paper and has not been reported previously in a published research paper, authors will be asked to fill out a Code and Software Submission Checklist1 and to provide all the required materials as listed below prior to peer-review."

Since this is an application of the model that has been reported previously in several publications including Nature journals, it seems that it is not subject to provide the checklist and code (e.g., Bottino et al., 2024; Sun et al., 2024).

In addition, while the guidelines may be tractable for small-scale software applications intended for a desktop computer, several aspects of the guidelines are intractable for global land surface models which are a component of Earth System Models and designed to run on hundreds to thousands of processors on a supercomputer (e.g., see Buechel, 2021). While there are efforts to develop "containers" for prominent earth system models that provide some platform transferability and could be distributed with major earth system model releases, such software remains nascent despite multi-year efforts. Adapting codes to alternative supercomputing platforms unfortunately still requires substantial technical support and platform-specific knowledge. The further burden of making global earth system models and their components

usable on a desktop is far beyond what the resources available for this contribution - an application of the previously published model - can bear.

As one of the most relevant examples, please see “How to use ORCHIDEE” at <https://orchidee.ipsl.fr/you-orchidee/>. It states that

“Downloading and running the code: Given the current limited engineering staff within the ORCHIDEE group, we can not provide the full support that would be needed to install and easily run the model on any server. In this context, please contact a relevant person from the ORCHIDEE project team (see list of the core team) to establish a more direct collaboration in order to benefit from the support/investment of an ORCHIDEE expert.”

ORCHIDEE is also a land surface model like LM3-TAN, and documentation/applications of its latest several versions have been published in Nature journals and elsewhere providing the full source codes like we have done (e.g., Beaudor et al., 2023; Nandintsetseg et al., 2024). To our knowledge, it has not been common to provide a how-to-run manual for any server for this type of models.

We have also endeavored to look Nature journal papers up to consult how global land surface models or Earth System Models could have provided code installation/software guide satisfying the “Guidelines for authors submitting code & software” of Nature journals. We, however, have not yet been able to find an example.

Given the points above, we feel that provision of the code and thorough documentation (both in this paper and those before it) is consistent with or exceeds journal requirements and current practice and capabilities for applications of previously published earth system modeling components such as ours. We are furthermore happy to engage with those interested in running our large-scale computationally-intensive model to help configure it for their systems.

We hope that this approach will be sufficient, and would welcome the opportunity to discuss if the editor or reviewer has questions or concerns.

Beaudor, M. et al. Global agricultural ammonia emissions simulated with the ORCHIDEE land surface model. *Geosci. Model Dev.* 16, 1053–1081 (2023). <https://doi.org/10.5194/gmd-16-1053-2023>

Bottino, M.J., Nobre, P., Giarolla, E. *et al.* Amazon savannization and climate change are projected to increase dry season length and temperature extremes over Brazil. *Sci Rep* 14, 5131 (2024). <https://doi.org/10.1038/s41598-024-55176-5>

Buechel, M. Understanding hydrological change with land surface models. *Nat Rev Earth Environ* 2, 824 (2021). <https://doi.org/10.1038/s43017-021-00241-0>

Sun, J., Wang, Y., Lee, T.M. *et al.* Nature-based Solutions can help restore degraded grasslands and increase carbon sequestration in the Tibetan Plateau. *Commun Earth Environ* 5, 154 (2024). <https://doi.org/10.1038/s43247-024-01330-w>

Nandintsetseg, B., Chang, J., Sen, O.L. *et al.* Future drought risk and adaptation of pastoralism in Eurasian rangelands. *npj Clim Atmos Sci* 7, 82 (2024). <https://doi.org/10.1038/s41612-024-00624-2>

File name/ Data description (Size)	Availability
hydrography.1deg_cm2.20131203.tile1.nc, geohydrology_table.20090108.nc/ Geohydrology information for the river component of the model	https://github.com/minjinl/LM3-TAN
cover_frac_m45.20080723.nc/ Land cover information	
ground_frac_m45.20080723.nc/ Soil type information	
navy_topography.data.nc/ Topographic data	
1deg_mosaic_file_from_zhi_20nov2013.cpio/ Description of the model grid at 1 by 1 degree resolution (34MB)	This input dataset totaling 362 MB is available from the corresponding author upon request.
sst_data.cpio/ Sea surface temperature (50MB)	
geohydrology.20090108.nc/ Geohydrology information for the river component of the model (64MB)	
surface_reflectance.20090108.nc/ Soil surface reflectance (214 MB)	
Climate forcing (See Methods)	This forcing dataset totaling 899 GB is available from the corresponding author upon request.
Atmospheric CO2 concentration (See Methods)	https://esgf-node.llnl.gov/search/input4mips/ , last access 3/2/2024
Land use and fertilizer applications (See Methods)	https://uh.umd.edu/data.shtml , last access 3/2/2024
Manure applications (See Methods)	https://zenodo.org/records/10100435
Atmospheric deposition (See Methods)	https://esgf-node.llnl.gov/search/input4mips/ , last access 3/2/2024

Table S7. Model input data description and availability.